# Homing gene drives can transfer rapidly between *Anopheles gambiae* strains with minimal carryover of flanking sequences

Poppy Pescod [1], Giulia Bevivino[2], Amalia Anthousi[1,3,4], Josephine Shepherd[1], Ruth Shelton [1], Fabrizio Lombardo [2] & Tony Nolan [1] ✉

CRISPR-Cas9 homing gene drives are designed to induce a targeted double-stranded DNA break at a wild type allele ('recipient'), which, when repaired by the host cell, is converted to the drive allele from the homologous ('donor') chromosome. Germline localisation of this process leads to super-Mendelian inheritance of the drive and the rapid spread of linked traits, offering a novel strategy for population control through the deliberate release of drive individuals. During the homology-based DNA repair, additional segments of the recipient chromosome may convert to match the donor, potentially impacting carrier fitness and strategy success. Using *Anopheles gambiae* strains with variations around the drive target site, here we assess the extent and nature of chromosomal conversion. We show both homing and meiotic drive contribute as mechanisms of inheritance bias. Additionally, over 80% of homing events resolve within 50 bp of the chromosomal break, enabling rapid gene drive transfer into locally-adapted genetic backgrounds.

Although significant progress has been made towards elimination of malaria in the last two decades, with an estimated 2 billion cases of malaria averted, progress has begun to slow[1]. An important cause of this deceleration is resistance, of the parasite to treatment and of the *Anopheles* mosquito vector to insecticides used in long-lasting insecticidal bednets[2]. The World Health Organization has stated that novel control strategies are required to maintain current gains and advance further towards malaria elimination, including genetic control strategies such as gene drive[3].

Broadly defined, a gene drive is any selfish element that promotes its own inheritance over and above natural Mendelian inheritance rates. As a result of this bias, gene drives can invade a population, even if they impose a fitness cost[4,5]. Such selfish genetic elements are abundant in nature, and their characteristics are being exploited in synthetic gene drives to tackle multiple issues including agricultural pest control, infectious disease transmission, and loss of biodiversity[5,6].

Gene drives can be used in two broad strategies: population modification or population suppression[7]. Population suppression strategies may target a haplosufficient gene (essential for development or reproduction, but capable of normal function with one intact copy), where insertion of the gene drive cassette produces a null allele. Alternatively, a population can be modified by introducing an effector gene capable of altering traits associated with vectorial capacity within the gene drive cassette as cargo[8]. *Anopheles gambiae*, the main malaria vector, appears to be particularly well suited to gene drive strategies; inheritance rates of gene drives in *An. gambiae* are commonly 97–100% in multiple target sites[9–12], higher than those routinely observed in *Drosophila* (~80%)[13] and *Aedes* (~70%)[14].

The mechanisms by which gene drives bias their inheritance are broad, but most gene drives currently under development emulate homing endonuclease genes (HEGs). HEGs are naturally-occurring selfish genes encoding enzymes that recognise and cut specific genomic target sequences[15]. A HEG is ordinarily located within its own genomic target sequence and cleaves any homologous chromosome which does not contain the HEG (and therefore has an intact target sequence). Target sequence cleavage induces the cell's own homology directed

[1]Vector Biology Department, Liverpool School of Tropical Medicine, Liverpool, UK. [2]Division of Parasitology, Department of Public Health and Infectious Diseases, University of Rome "la Sapienza", Rome, Italy. [3]Department of Biology, University of Crete, Vassilika Vouton, Heraklion, Crete, Greece. [4]Institute of Molecular Biology and Biotechnology, Foundation for Research and Technology-Hellas, Heraklion, Greece. ✉e-mail: Tony.Nolan@lstmed.ac.uk

repair system, using the homologous chromosome containing the HEG gene as a repair template, which results in copying of the HEG to the repaired chromosome. Thus, germline cells undergoing this process become homozygous for the HEG so that most progeny will inherit it[15,16]. Use of HEGs was proposed as a tool for population-wide genetic engineering in 2003[7] but only modest progress was made due to inherent limitations in available genomic target sites and difficulties in their molecular engineering, until the discovery of the CRISPR system[8].

In principle, any nuclease which can be redirected to a specific target site of choice could potentially be used as a HEG in a synthetic gene drive system. CRISPR-associated protein 9 (Cas9), when associated with an easily reprogrammable 20 nt guide RNA (gRNA), can produce a double-stranded break (DSB) at almost any target sequence – the only limit is the requirement for a small protospacer-adjacent motif immediately following the sequence complementary to the gRNA[17]. Genes encoding for the Cas9 and the appropriate gRNA are inserted into one chromosome as a gene drive cassette and expressed to produce a Cas9-gRNA complex, which targets and cuts the homologous chromosome to insert the cassette at homozygosity in a process known as homing[18]. A single drive allele can be inherited by up to 100% of offspring, depending on the level and timing of Cas9 expression in the germline.

CRISPR-based homing gene drives expressed in the germline have been developed in multiple mosquito species and have demonstrated that *Anopheles sp.* are well suited to gene drive control strategies, with rates of inheritance bias close to 100%[9–12]. However, potential barriers to their success in vector control strategies have been identified. A common concern is whether the lab-bred mosquito strains containing the gene drive will be capable of rapid transfer into wild populations due to lack of fitness after years of lab maintenance, slowing the initial spread of the gene drive element into the target population. Once the gene drive has spread into the target population it may face the development of resistance due to standing resistance caused by natural variation around the target site, or CRISPR-mediated cut and repair errors, but there are a number of strategies being developed to mitigate this[11,19–22].

As well as biasing inheritance by homing, gene drives are capable of super-Mendelian propagation by excluding the paired chromosome not containing the gene drive cassette, eliminating wild type gametes in a process known as meiotic drive. This can be a deliberate feature of a gene drive, such as chromosome-shredder meiotic drives[23,24], and recently it has been proposed as an unintended and context-dependent outcome of homing-based gene drives[25]. The phenotypic effects of the two drive mechanisms – increased inheritance of the gene drive element by progeny – are largely indistinguishable from one another but may have consequences for the spread or suppression of non-target regions by their linkage to the gene drive.

CRISPR-Cas9 edits a gene via the induction and repair of a DSB, using the endogenous DNA repair mechanism of the host cell. Several highly conserved pathways of repair for DSBs are grouped into two general strategies: end joining, which does not require a template and causes small insertions or deletions (indels) at the cut site; and homology-directed repair (HDR). HDR requires a template homologous to the regions either side of the DSB; repair by synthesis from the intact template containing the gene drive allele leads to copying of the allele onto the repaired chromosome.

HDR events are initiated by resection of the 5' ends at the DSB to produce single stranded 3' overhangs, followed by the ssDNA search for a homologous template to copy, and end with ligation of the repaired ends[26]. As the donor is used as a template for the DSB repair, any variation present in the donor sequence compared to the recipient will be copied into the recipient, converting any heterozygous regions to homozygosity. The length of resection of the 5' DSB ends determines the length of donor sequence copied into the recipient – producing so-called gene conversion tracts (GCTs) of varying length in the recipient chromosome. This may influence the performance of

synthetic gene drives when released into target populations in the field; despite rigorous selection of conserved target regions with little variation, regions around gene drive target sites are likely to vary in diverse wild populations. While a gene conversion around a gene drive target site may involve a small fraction of the overall genome, there are potential phenotypic impacts of loss of heterozygosity in the surrounding sequence. Moreover, the amount of additional genomic sequence that accompanies any homing event from its lab-selected genome into a field-selected genome will be an important question for regulators.

Studies of GCTs formed after HEG-mediated DSBs are limited in most organisms and we are unaware of any studies in *An. gambiae*, despite its importance for public health and the successful development of several gene drive strategies in the lab. CRISPR-Cas9 mediated DSBs have been seen to induce GCTs of up to 1.7 kb in yeasts[27,28]. I-Sce1 mediated DSBs have induced GCTs of up to 0.5 kb in mammalian stem cells[29], and up to 0.8 kb (average 471 bp) in *Drosophila melanogaster*[30,31]. In mosquitoes, GCTs of up to 700 bp have been observed in *Aedes aegypti* – however, these involved HDR between the cut chromosome and a plasmid donor template, rather than between homologous chromosomes, and therefore may not be representative of gene drive repair dynamics[32].

Recent work has shown that *An. gambiae* gene drives are remarkably robust to sequence heterology around the target locus, with indistinguishable homing rates between strains with up to a 6.6% sequence difference[33], while in one study a difference of 1.2% in *Ae. aegypti* was sufficient to cause a 66% reduction in successful repair of a Cas9-mediated knock-in[32]. These results imply different DNA repair dynamics in both species, making it likely that there may also be differences in GCTs from gene drive homing in *An. gambiae* compared to other Diptera. The ability of *An. gambiae* gene drives to home into regions of heterology also raises the possibility that adjacent SNPs in the donor chromosome may be transferred along with the gene drive. This could cause the inadvertent linkage of surrounding sequences to the gene drive, spreading them into a population regardless of fitness cost. While this is unlikely to have a phenotypic effect, particularly in population eradication strategies, being able to predict any potential changes to the target population may help ensure the success of the control strategy. Therefore, for any gene drive designed for release into wild heterogeneous populations of *An. gambiae*, it is important to understand the extent of genetic transfer between donor and recipient chromosomes during gene drive homing.

In this study we exploit natural variation around two gene drive target sites, between the lab reared "G3" *An. gambiae/coluzzii* hybrid strain in which the gene drives were designed and a more recently caught 'wild' *An. coluzzi* strain "N'Gousso", to determine the extent of genetic transfer during gene drive homing and to confirm the mechanism of inheritance bias.

## Results

### Resolution of SNPs for chromosomal conversion analysis

We set out to examine the amount of gene conversion that occurs during the homing of a gene drive from one chromosome to another, using two gene drive systems targeting different genes associated with female fertility: *zpg*−7280 targeting *nudel* (AGAP007280) and *vas2*−5958 targeting *yellow-g* (AGAP005958)[11,21]. We created $F_1$ hybrid mosquitoes between each of these gene drive strains, which were both generated in a G3 strain genetic background, and a more recently colonised wild type (N'Gousso) strain[34]. SNPs that were private either to the G3 chromosome or the N'Gousso chromosome in the parents allowed us to distinguish the original gene drive donor chromosome from the 'homed' recipient chromosome in their offspring (Fig. 1). Gene drive inheritance rates were between 87 and 100% in the progeny of the $F_1$ hybrids, which is within normal range of inheritance values for both gene drives[11,21].

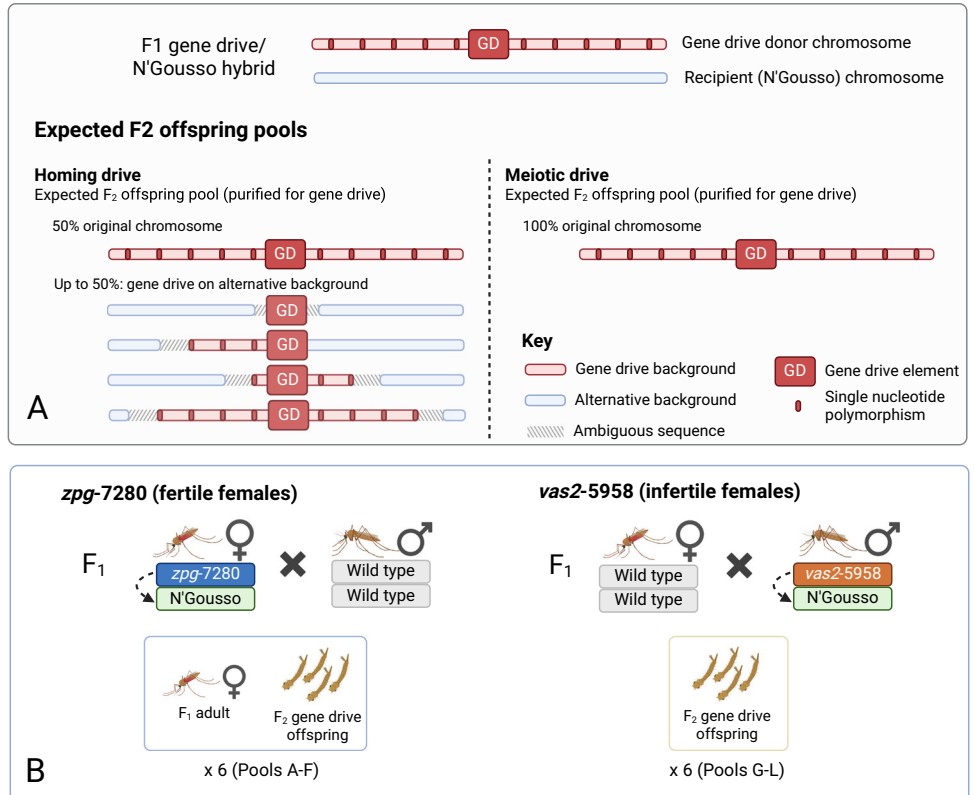

**Fig. 1 | Expected gene drive haplotypes among offspring from an F₁ gene drive/wild type strain hybrid, according to the mechanism of biased inheritance.** **A** Single nucleotide polymorphisms (SNPs) around the gene drive site between the F₁ parent chromosomes allow identification of the parent chromosome inherited by each offspring, as well as the presence and amount of resection produced during gene drive homing via SNP presence or absence. Regions between present and absent SNPs are labelled ambiguous as there is no way to determine the origin of these sequences, or where conversion tracts produced during homing end within these sequences. An absence of any offspring containing the gene drive element on the alternate background indicates inheritance is biased by a meiotic drive mechanism, whereby the alternate chromosome is cut and destroyed. **B** Crosses of F1 hybrids used to produce offspring pools: Pools A-F were collected from *zpg*−7280/N'Gousso hybrids, with the female parent of each pool also collected. Pools G-L were collected from individual WT females mated *en masse* to *vas2*−5958/N'Gousso hybrid males (since females of this line are sterile); therefore, samples of the gene drive parent could not be taken. Figure 1 was created with BioRender.com released under a Creative Commons Attribution-NonCommercial-NoDerivs 4.0 International license (https://creativecommons.org/licenses/by-nc-nd/4.0/deed.en).

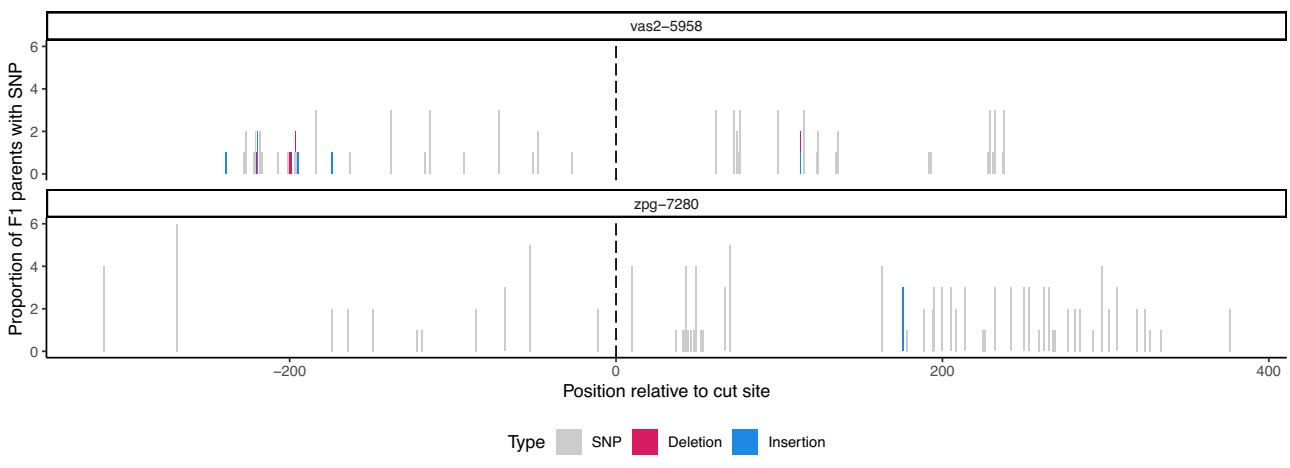

**Fig. 2 | Heterology around the gene drive sites.** Single nucleotide polymorphisms (SNPs) and small insertions or deletions between the chromosomes of F₁ hybrids (*vas2*−5958/N'Gousso or *zpg*−7280/N'Gousso) in a window of 400 bp either side of the gene drive insertion, with positions given relative to the gene drive cut site. In each cross, SNPs private to one chromosome were informative for determining gene conversion tract length. Source data are provided as a Source Data file.

Six pools of F₂ offspring from both gene drives and, where available, their parents (Fig. 1B) were retained for sequencing to determine the degree of carryover of the sequences surrounding the gene drive into the recipient chromosome (offspring pools A-F from *zpg*−7280, offspring pools G-L from *vas2*−5958). The limit of resolution for our assay to determine GCT length is dependent on the frequency and distribution of SNPs that distinguish the donor chromosome from the recipient chromosome in the ~800 bp region surrounding the gene drive insertion site (Fig. 2). In the case of the *zpg*−7280 gene drive, sequencing the F₁ hybrid parents across the 6 pools revealed on

average there was 3.6% difference between donor and recipient chromosomes, with an average 22 (14–34) nucleotide locations containing a SNP. In the *vas2*–5958/N'Gousso pools we were unable to sequence the parents, but by using the pool sequences (see Methods) we determined that there was 2.5% difference between donor and recipient chromosomes, with an average 14 (0–23) nucleotides differing between the chromosomes in the sequenced region. In two vas2-5958/N'Gousso pools (I and J) no nucleotides differing between the parent chromosomes were discovered; these pools were removed from this analysis.

The distribution of SNPs between the donor and recipient chromosomes gave an average range of 34–319 bp from the cut site in the *zpg*–7280/N'Gousso crosses and 48–224 bp in the *vas2*–5958/N'Gousso crosses. Of particular importance are the SNPs closest to the cut site, as the limiting factor to the resolution of small GCTs – for example, if the closest SNP to the cut site was 10 bp away, then any GCT shorter than 10 bp would be indistinguishable from a 'perfect' homing event with no GCT. The distance of the closest SNP to the cut site on either side was 11–71 bp in *zpg*–7280/N'Gousso hybrids and 27–63 bp in *vas2*–5958/N'Gousso hybrids.

## Gene drive homing can produce small gene conversion tracts

In each pool of offspring from gene drive/N'Gousso hybrids, we found both the original donor haplotype of the gene drive line and multiple haplotypes consistent with gene drive homing onto the recipient chromosome, with varying amounts of gene conversion (Figs. 3–5). Amplicons from either side of the gene drive target site in pooled offspring of each $F_1$ hybrid parent were sequenced to determine the frequency and gene conversion content of homed haplotypes. Sequences were grouped into haplotypes based on whether they matched the gene drive donor chromosome or (at least partially) the recipient chromosome. We split the recipient haplotypes into major (≥10% of filtered alleles) and minor groups (<10%), since in the latter category it was difficult to distinguish low frequency haplotypes consistent with gene drive homing from PCR artifacts (see Supplementary Information 2).

Homing events with no carryover of sequences from the donor chromosome accounted for 12/22 of the major recipient haplotypes and 48% of all recipient haplotype alleles, including minor alleles. Six out of 22 of the major recipient haplotypes contained only one SNP from the donor sequence, and the remaining four haplotypes had 2–5 SNPs carried over. The furthest SNP from the gene drive that was carried over as part of a GCT in a major haplotype was 86 bp away from the cut site (Pools A and D). Minor haplotypes made up 44/86 of haplotypes and 8.0% of reads; most represented longer GCTs, with 21/44 of minor haplotypes (2.5% of reads) showing GCTs of at least 100 bp from the gene drive cut site. The precise length of the GCTs cannot be identified by SNP presence/absence alone due to regions of uncertainty between present and absent SNPs, making the maximum potential GCT length in the major haplotypes 286 bp from the gene drive (Pool A, Fig. 3). However, overall 82.3% of all homing events (major and minor) were resolved within 50 bp of the gene drive site.

Long-read sequencing revealed a previously undocumented SNP in the primer sequence used to amplify the right-hand side of the cut site on the donor chromosomes of three *zpg*–7280 pools, skewing the proportions of donor and recipient haplotypes sequenced. These pools (A, B and D); the haplotypes and their skewed proportions are nonetheless shown in Figs. 3, 4, with impacted haplotypes indicated with an asterisk (*).

## Homing and meiotic drive both contribute to inheritance bias

In theory there are two possible ways that a nuclease-based gene drive of the type tested here could bias its inheritance: the classical HDR pathway of the cut chromosome using the gene drive-bearing chromosome as template, which leads to 'homing'; alternatively, the wild type chromosome may be selectively removed ('meiotic drive') during gametogenesis, following its cleavage by the nuclease. Some recent evidence suggests that both mechanisms can occur: mixed method gene drive inheritance has been postulated in *Ae. aegypti*, where context-dependent differences in DNA repair appeared to cause meiotic drive in some individuals and homing-based drive in others[25]. Our experimental setup, with naturally varying donor and recipient chromosomes, allows us to identify and quantify each outcome by looking at the nature and frequency of chromosomes among the offspring of heterozygous gene drive parents; presence of the gene drive on the recipient chromosome is indicative of homing, whereas an over-representation of the gene drive on the donor chromosome could be indicative of meiotic drive.

If meiotic drive were occurring in addition to homing, we should observe a higher-than-expected frequency of offspring with the gene drive on the donor (G3) chromosome. Because our assay looked only at individuals carrying the gene drive element, the expected frequency of donor haplotypes must be expressed as a percentage of those that contained the drive element, rather than the total offspring. Given the rate of inheritance of the gene drive was 87–100%, the expected frequency of offspring with the gene drive on a donor haplotype in each pool is slightly higher than 50%. For example, if there were 95% inheritance of the gene drive in a pool of offspring we would expect 53% (50/95) of sequenced offspring to contain the original donor haplotype in the absence of meiotic drive.

The expected proportion of haplotypes matching the gene drive donor chromosome was 50–58% in offspring pools from both gene drives. In the *zpg*–7280 gene drive we found the gene drive element in haplotypes matching the donor chromosome in 73% (58–85%) of all reads from the $F_2$ offspring pools, excluding those where a SNP was present in the primer sequence (Figs. 3, 4, 6). This was a significant difference between the observed proportions of homed (gene drive on recipient chromosome) haplotypes and the proportions expected if homing was the only source of inheritance bias (Pools C, E and F, paired t-test: $t = 5.25$, df = 2, $p = 0.03$), with significantly higher proportions of donor haplotypes than expected in each.

In offspring pools of *vas2*–5958/N'Gousso $F_1$ hybrids the most common haplotype in each pool was assumed to be the gene drive donor, which was found in 63% (48–84%) of reads. Of note is the large disparity in reads originating from the donor chromosome between the left and right side of the cut site in Pool L (*vas2*–5958/N'Gousso), with a frequency of 50% on the left and 82% on the right (Figs. 5, 6). It is unlikely that this is a true biological representation of the haplotype frequencies in this pool; more likely this is due to an unknown SNP or a difference in binding inefficiency in the primers for the left side of the cut site in the same way as seen in Pools A, B and D, and the true donor sequence proportion is closer to that seen on the right side. With Pool L removed from the *vas2*–5958 data, the difference between observed and expected donor haplotypes was not significant in offspring of the remaining *vas2*-5958/N'Gousso hybrids (Pools G, H and K, paired t-test: $t = 0.95$, df = 2, $p = 0.44$). However, by looking at the haplotype proportions in each sample it is clear that Pool G has a higher representation of donor chromosomes (78–84%) than Pools H and K (51–54% and 48–55% respectively) (Fig. 6). Additionally, if the proportion of haplotypes matching the gene drive donor in Pool L is likely to be closest to the right hand, Pool L would also have a higher donor representation than expected based on homing alone (84%).

An alternative explanation for the observation of over-representation of chromosomes with the gene drive on an apparent donor haplotype would be that there is simply a longer resection, leading to a longer GCT on the recipient chromosome than our sequencing window can capture (~400 bp), making the sequence indistinguishable from a donor haplotype. To rule this out, longer 4 kb amplicons flanking the gene drive on each side were sequenced from the *zpg*–7280 drive $F_1$ hybrids and offspring. Due to the extended

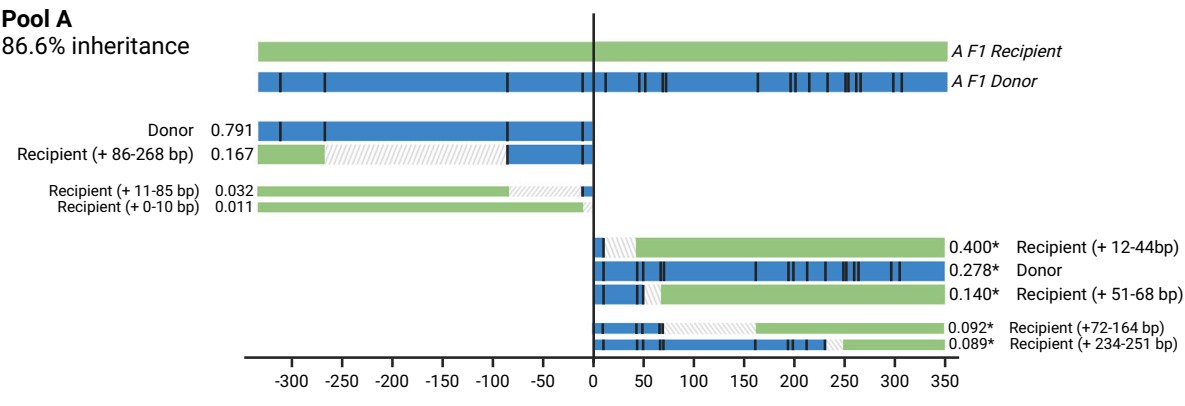

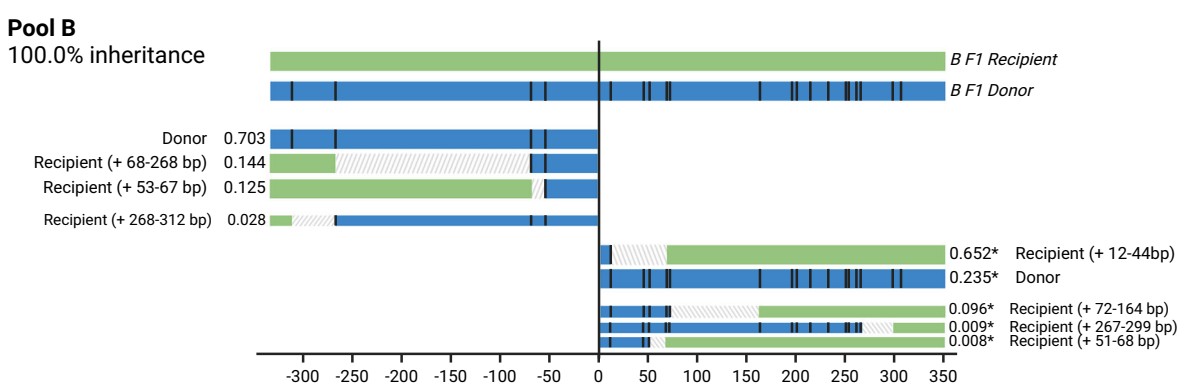

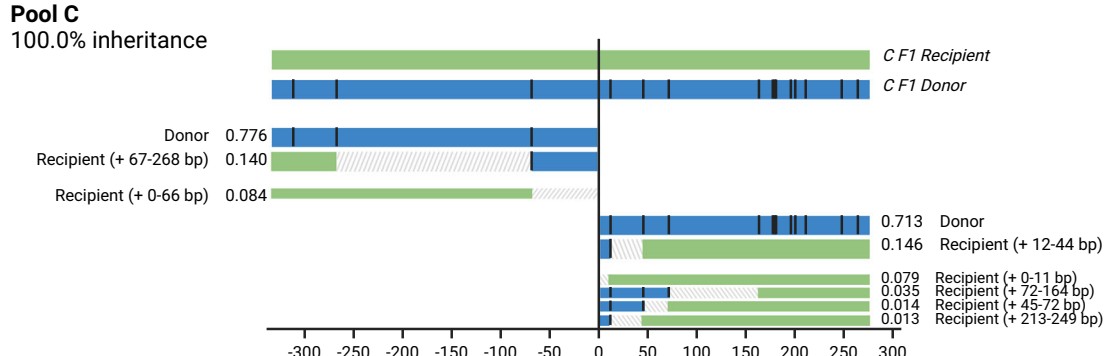

**Fig. 3 | Offspring haplotypes from *zpg*−7280/N'Gousso hybrid females, pools A-C.** Haplotypes of offspring from parents A-C (see Supplementary Table 1) showing gene conversion tracts (GCTs) generated during gene drive homing based on presence and absence of SNPs present between the parent chromosomes (represented above each graph). The relative proportion of each haplotype is given as well as a description of the haplotype. Black lines represent a SNP present in the gene drive donor (G3) chromosome of the parent and absent in the recipient (N'Gousso) chromosome. Regions between a present and absent SNP are labelled as ambiguous, as gene conversion may have ended at any point in this region. Alleles present at <10% relative abundance after filtering are shown with narrow bars, as they may be either a sequencing artifact or a minor homed allele. *proportions are inaccurate due to the presence of a SNP in the binding site of the primer used to amplify the genomic sequence on the donor chromosome. Figure 3 was created with BioRender.com released under a Creative Commons Attribution-NonCommercial-NoDerivs 4.0 International license (https://creativecommons.org/licenses/by-nc-nd/4.0/deed.en).

length of the amplicon, our sequencing approach did not permit the recovery of individual haplotypes. Instead, we compared sequences from each offspring pool to the gene drive donor chromosome of their parent, to calculate the proportion of reads at each nucleotide in the offspring sequences that originated from a gene drive recipient chromosome (Fig. 7). By plotting the frequency of recipient SNPs at each position along the sequence it was possible to visualise any resection events as slopes in the trend of variant alleles; a resection event would show as an increase in variant SNP frequency as the position moves away from the cut site. The known small resections within the original short read sequence can be seen (Fig. 7 and Supplementary Fig. 10) with no obvious additional slopes to suggest long resections that might otherwise cause the increased abundance of apparent donor haplotypes in the short read pools. Therefore, meiotic drive is the most parsimonious explanation for the over-abundance of gene drive donor haplotypes that we observed. Similarly to Pools A, B, D and L in the short-read experiment, Pool F appears to have a slightly lower frequency of reads matching the donor chromosome on the left of the cut site than the right. This may be due to a similar issue with primer binding or the difference in read depth between the left and

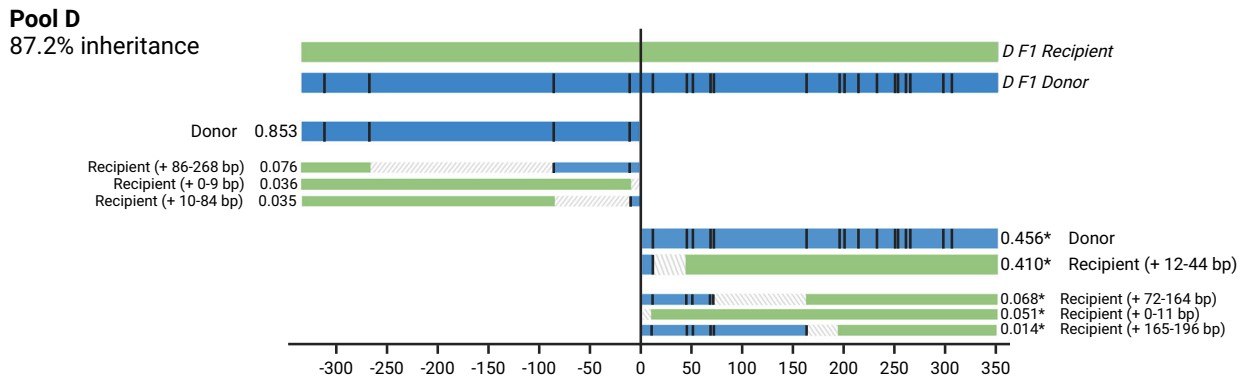

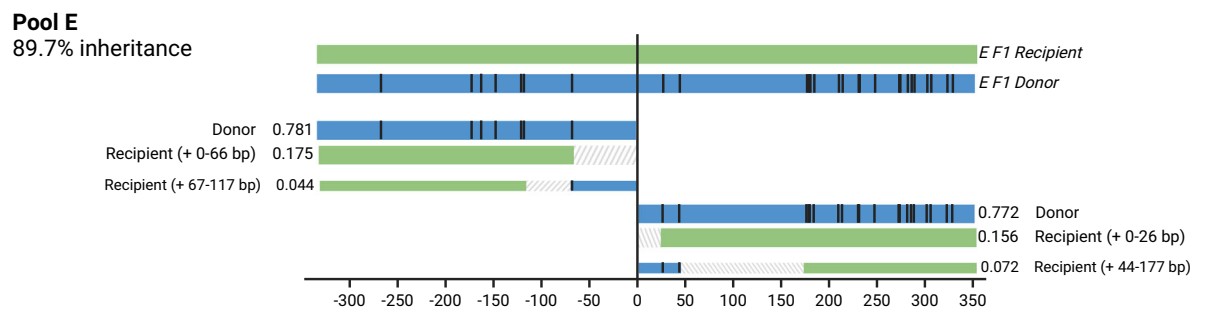

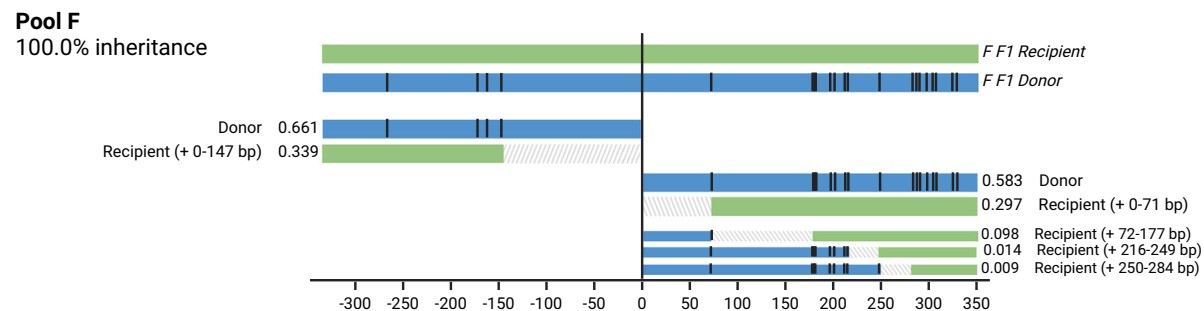

**Fig. 4 | Offspring haplotypes from *zpg*−7280/N'Gousso hybrid females, pools D-F.** Haplotypes of offspring from parents D-F (see Supplementary Table 1) showing gene conversion tracts (GCTs) generated during gene drive homing based on presence and absence of SNPs present between the parent chromosomes (represented above each graph). The relative proportion of each haplotype is given as well as a description of the haplotype. Black lines represent a SNP present in the gene drive donor (G3) chromosome of the parent and absent in the recipient (N'Gousso) chromosome. Regions between a present and absent SNP are labelled as ambiguous, as gene conversion may have ended at any point in this region. Alleles present at <10% relative abundance after filtering are shown with narrow bars, as they may be either a sequencing artifact or a minor homed allele. *proportions are inaccurate due to the presence of a SNP in the binding site of the primer used to amplify the genomic sequence on the donor chromosome. Figure 4 was created with BioRender.com released under a Creative Commons Attribution-NonCommercial-NoDerivs 4.0 International license (https://creativecommons.org/licenses/by-nc-nd/4.0/deed.en).

right sides (Supplementary Figs. 7, 8), but does not impact the determination of resection length using the changes in slope.

## Discussion

Gene drives are generated in a lab environment, within lab-reared strains that have previously been isolated from wild populations, often decades ago. Additionally, *An. gambiae* has a uniquely diverse genetic background, with an estimated one variant allele every 1.9 bases of the accessible genome[35], as well as complex introgression dynamics within the interbreeding *An. gambiae* complex[36]. In a diverse species complex like *An. gambiae* it is important to understand the dynamics of gene transfer between donor and recipient genetic backgrounds, in advance of releases for control of wild populations. The efficient transmission of a field-released gene drive to target populations could be compromised by poor fitness and mating competitiveness of lab-bred mosquito strains, or by their offspring containing large conversion tracts

from a lab-bred strain. For example, lab-bred strains may be less adapted to the local ecology and conditions or may have a different insecticide resistance status than wild strains in the release area. In this study we demonstrate that a gene drive is capable of transferring rapidly from its lab-bred genetic background into a wild strain, with minimal or no gene conversion. This would prevent chromosomes from the lab strain impacting fitness while the gene drive element spreads into the field population.

The lack of large conversion tracts between chromosomes during homing reduces the potential of gene drives inadvertently spreading flanking alleles into target populations, which would be inherited at a rate above Mendelian inheritance. The inheritance rate of surrounding sequences correlates with their proximity to the gene drive construct – closer sequences are more likely to be carried over during homing. In this study we have showed that the majority of homing events cause minimal GCTs in the recipient chromosome; in major alleles, 87% of

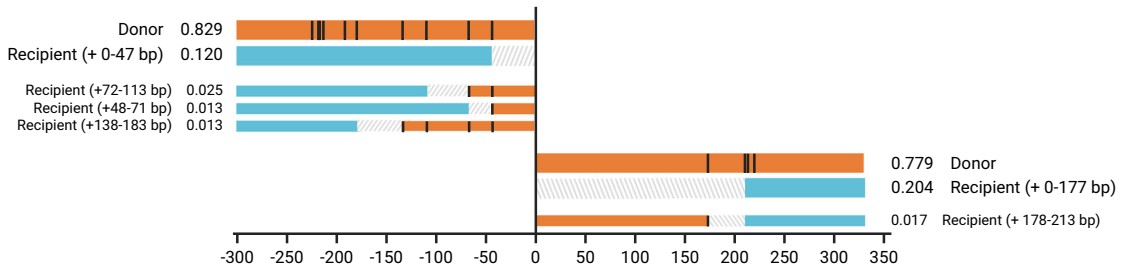

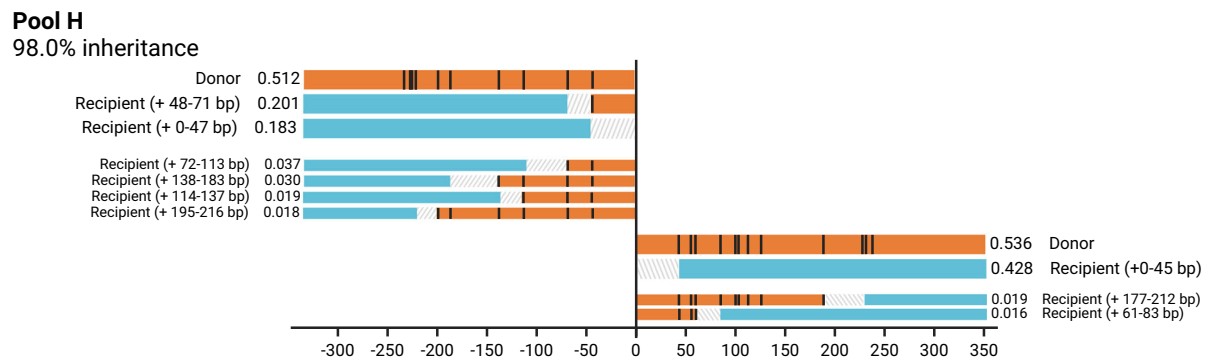

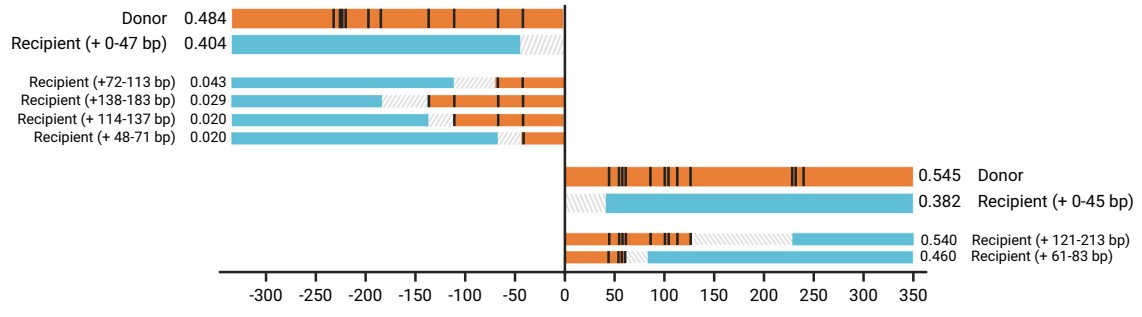

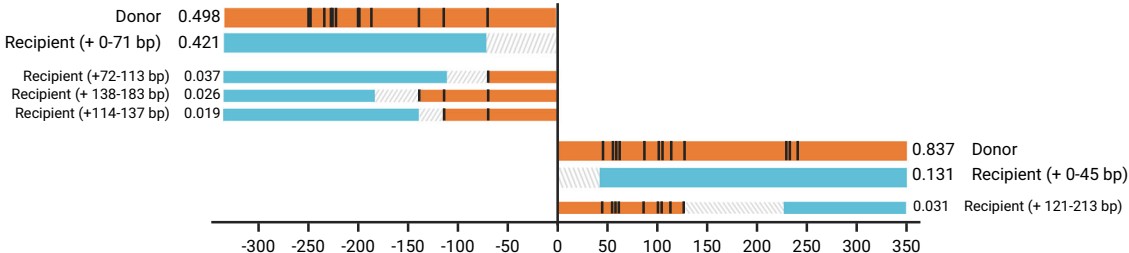

**Fig. 5 | Offspring haplotypes from *vas2*−5958/N'Gousso hybrid parents, pools G-L.** Haplotypes of offspring from parents G, H, K and L (see Supplementary Table 1) showing gene conversion tracts (GCTs) generated during gene drive homing. Due to the use of males mated *en masse* in *vas2*−5958 crosses the gene drive parent could not be sequenced; therefore, classification of haplotypes as from donor or recipient chromosomes are based on the assumption that the most abundant haplotype in each *vas2*-5958 matches the donor chromosome of the parent. Black lines represent a SNP present in the donor (G3) chromosome and absent in the recipient (N'Gousso) chromosome. Regions between a present and absent SNP are labelled as ambiguous, as gene conversion may have ended at any point in this region. Alleles present at <10% relative abundance after filtering are shown with narrow bars, as they may be either a sequencing artifact or a minor homed allele. Figure 5 was created with BioRender.com released under a Creative Commons Attribution-NonCommercial-NoDerivs 4.0 International license (https://creativecommons.org/licenses/by-nc-nd/4.0/deed.en).

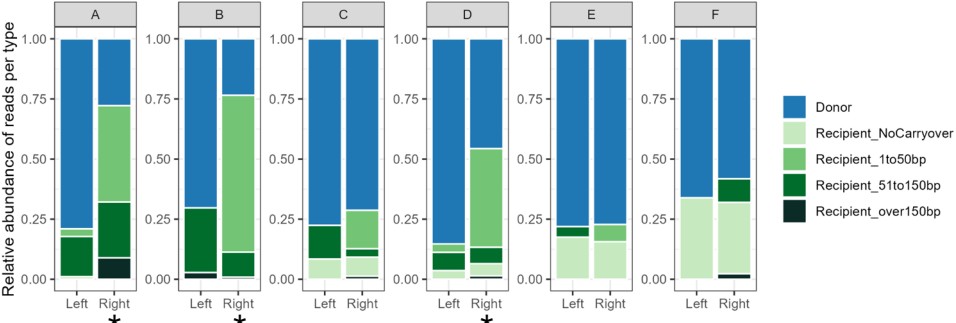

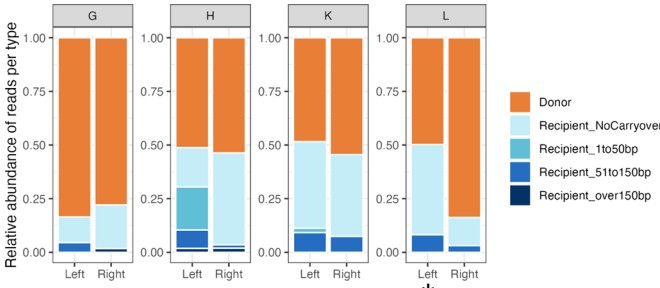

**Fig. 6 | Relative abundances of gene conversion tract haplotypes in each pool.** Haplotypes are classified by the position of the closest SNP to the gene drive cut site matching the donor chromosome on a recipient chromosome background. *Samples in which proportions of the donor sequence are artificially reduced by a SNP in the primer sequence (confirmed by long-read sequencing in *zpg*−7280/

N'Gousso samples). Figure 6 was created with BioRender.com released under a Creative Commons Attribution-NonCommercial-NoDerivs 4.0 International license (https://creativecommons.org/licenses/by-nc-nd/4.0/deed.en). Source data are provided as a Source Data file.

reads from homed haplotypes contained no SNPs from the donor, no more than 5 adjacent SNPs were copied into the recipient chromosome, and the maximum observed potential GCT length was 286 bp. Longer potential GCTs were only present in low abundance seen in the minor alleles, which were indistinguishable from PCR artefacts. While in most cases small GCTs during homing are unlikely to be an issue, as newer gene drive targets are being specifically chosen for their low variability to reduce off-target binding and resistance development, it should be considered during gene drive design and regulation. It may also be considered during design of other CRISPR-based manipulations of *Anopheles* mosquitoes; for example, shorter GCTs would need to be taken into account if implementing a toxin-antidote recessive embryo drive, which requires resection to copy over a recoded resistant gRNA target site[37]. These short GCTs could also be used positively – for example, for super-Mendelian correction of an undesirable allele by producing a gene drive in a lab strain immediately adjacent to a desirable version of the allele and releasing it into a field population where gene conversion would carry over the desired allele with the gene drive, albeit at low efficiency.

While the predominant method of inheritance bias in the two gene drives studied was homing of the gene drive element into a recipient chromosome, there was also consistent evidence to suggest that there was a reduction in inheritance of the recipient chromosome in the *zpg*−7280 gene drive line and in some pools of the *vas2*−5958 line. A previous study described a system in *Aedes* where a Cas9-based gene drive had biased inheritance in males caused entirely by meiotic drive, although this proved irreproducible[25,38], suggesting that inheritance bias mechanisms may be context dependent. In this study we suggest that, rather than entirely by homing or by meiotic drive, mixed methods of inheritance bias may occur simultaneously in the same

gene drive individual. The mechanism leading to meiotic drive is not yet determined but may relate to selective removal of gametes containing a nuclease-cleaved chromosome, or a general fitness reduction or slower maturation of these gametes.

Understanding the detailed mechanisms underpinning gene drive should inform regulatory and stakeholder decision-making for its potential use in the field. In this work we have shed light on what kind of homing events can be expected and we conclude that inheritance for these two gene drives is predominantly biased by homing of the construct into the recipient chromosome, with some additional bias attributable to meiotic drive. We have also showed that the majority of homing events involve either no gene conversion either side of the gene drive or small amounts of gene conversion from donor to recipient, and that at least 82% of homing events are resolved within 50 bp of the cut site. Conclusions from this study have been strengthened by the use of two different gene drive strains, with different promoters driving Cas9 expression and gRNAs recognising different genomic target sites, and with homing examined in males (*vas2*−5958) or females (*zpg*−7280). The haplotype structures of offspring pools from both gene drive hybrids were similar; this suggests that, at least for target sites with similar amounts of variation to these two drives, the repair outcomes underpinning gene drive inheritance bias in *An. gambiae* are broadly predictable.

*An. gambiae* has consistently showed amenability to gene drive control, displaying higher than average inheritance rates when compared to similar species[9–14] and in the face of significant variation around the target site[33]. Our demonstration of the minimal amounts of gene conversion that occur during homing supports the idea that gene drives are able to fully integrate into new haplotypes from the first generation of hybrids between the released strain and target

### Pool C
Donor haplotype frequency: 0.738

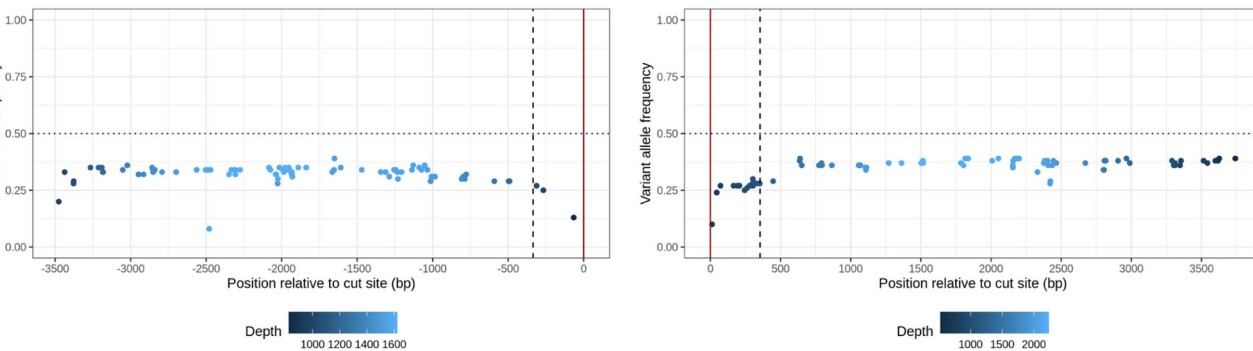

### Pool E
Donor haplotype frequency: 0.825

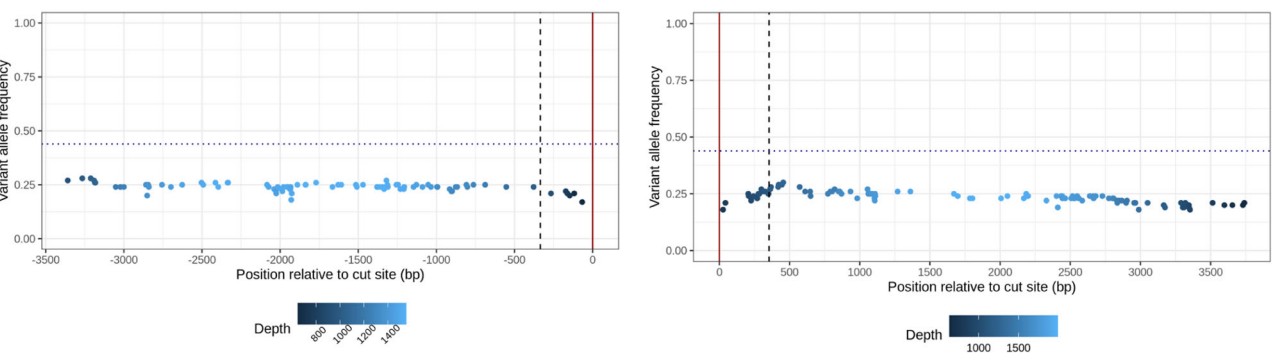

### Pool F
Donor haplotype frequency: 0.664

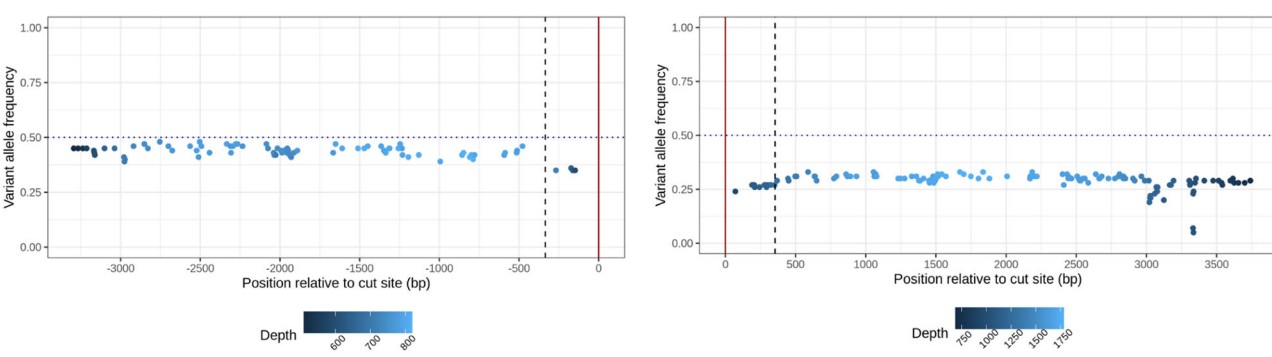

**Fig. 7 | Variant allele frequency in three _zpg_–7280/N'Gousso hybrid F₂ offspring pools 4 kb left and right of the gene drive insert site.** Frequency of variant alleles indicates the proportion of bases at each position which originated from the gene drive recipient chromosome in the parent. The horizontal dotted line shows the expected proportion of variant alleles (which represent the recipient haplotype) in each pool if all inheritance bias was due to gene drive homing rather than meiotic drive. The vertical dashed line shows the limit of the short-read sequencing done before, with a vertical red line indicating the gene drive cut site. Source data are provided as a Source Data file.

populations. The unexpected contribution of meiotic drive to inheritance rates in these gene drives may decrease the speed of this introgression; however, the homed gene drive haplotypes on local genetic backgrounds would likely be fitter in the field and therefore outcompete the original gene drive background, reducing the requirement of lab-bred genetic backgrounds to be competitive beyond the released generation.

## Methods
We confirm that this research complies with all relevant ethical regulations and did not need specific ethical approval.

### Mosquito rearing and strains
All mosquitoes were kept in standard rearing conditions of 26 ± 2 °C and 70 ± 10% relative humidity, with a 12 h light/dark cycle including hour long dusk/dawns. Larval stages were fed powdered fish food (TetraMin®), and adults were fed with a 10% sucrose solution *ad libitum*. Both gene drive lines had 30–40 G3 females added at each generation for maintenance. Adults were mated for 3–10 days before blood feeding for egg collection. Gene drive screening was performed on larvae or pupae by fluorescent light microscopy; both gene drive strains contain a red fluorescent protein marker expressed in the eyes[11]. The wild type and gene drive strains used are shown in Table 1.

## Mosquito crosses

Gene drive mosquitoes from the *vas2*–5958 and *zpg*–7280 lines were crossed to wild type N'Gousso to produce F$_1$ hybrid offspring containing one G3 chromosome with the gene drive cassette and one chromosome of N'Gousso with the wild type allele. These were screened for the gene drive element and positive individuals were backcrossed to G3. Both gene drive lines target female fertility genes but with different Cas9 promoters (Table 1); *vas2* expression is not entirely contained within the germline, and therefore shows some somatic expression of this gene drive resulting in female sterility. For this reason, in *zpg*–7280 gene drive crosses females could be used as the gene drive parent whereas only males can be used to propagate the *vas2*–5958 gene drive (11). Therefore, for *zpg*–7280 crosses the F$_1$ gene drive female parent was forced to lay singly, and F$_1$ parents were kept for sequencing alongside pools of their larval offspring. For *vas2*–5958 crosses, wild type G3 females mated to F$_1$ hybrid males were forced to lay singly, and pools of larval offspring from each single unknown F$_1$ parent were kept for sequencing. Females were assumed to have only mated with one male, as is the case in 85–88% of *An. gambiae* females in caged conditions[39,40]. Six pools from each gene drive/N'Gousso F$_1$ hybrid were analysed; details of crosses can be seen in Table 2. All gene drive positive F$_2$ offspring contained either the original G3 donor chromosome or an N'Gousso chromosome with a homed copy of the gene drive, as well as a wild type G3 chromosome which was not analysed.

## DNA extraction and sequencing

DNA was extracted from F$_1$ hybrid parents of *zpg*–7280 crosses for Sanger sequencing using the Livak method[41]; flies were homogenised in Livak buffer (0.5% SDS, 0.08 M NaCl, 0.16 M sucrose, 0.06 M EDTA, 0.12 M Tris-HCl, pH 9) and incubated at 65 °C for 30 min before precipitation with potassium acetate and incubation on ice for 30 min. Samples were centrifuged for 10 min at max speed and DNA was precipitated and pelleted using absolute ethanol. DNA pellets were washed with cold 70% ethanol and dissolved in molecular grade water. DNA extractions of F$_2$ offspring for Illumina sequencing were performed on pools of F$_2$ larvae from each parent using the Wizard® Genomic DNA extraction kit (Promega, Cat. # A2920). Primer pairs were designed to amplify ~400 bp either side of the cut site on the wild type and gene drive chromosomes (Supplementary Fig. 11); primer

sequences can be found in Supplementary Table 2. Wild type (N'Gousso) and gene drive (G3) chromosomes from *zpg*–7280 F$_1$ hybrids were amplified using these primers either side of the cut site, and chromosomes containing the gene drive were amplified from *zpg*–7280 and *vas2*–5958 F$_2$ pools. All PCR reactions were performed using Phusion Hot Start II High-Fidelity DNA polymerase (Thermo Scientific™, Cat. # F549S), 0.5 μM of each primer, and 1.8–6 ng/μl DNA in a 50 μl reaction. Cycle conditions were: 98 °C for 30 s, 30 cycles of 98 °C for 30 s, a variable annealing temperature (Supplementary Table 3) for 30 s, and 72 °C for 15 s, followed by a final extension step of 72 °C for 10 min. PCR products were purified using a QIAquick PCR Purification kit (QIAGEN, Cat. # 28104) and checked for purity using a DNA spectrophotometer.

Amplified gene drive and wild type chromosome fragments either side of the cut site from the F$_1$ hybrid parents of *zpg*–7280 crosses were subjected to bidirectional Sanger sequencing (GENEWIZ). Target locus heterology – the proportion of nucleotides different between the parent chromosomes – was calculated as a percentage of the total sequence length for each parent, excluding the gene drive cassette. Amplicons from gene drive-containing chromosomes in F$_2$ larval pools were sequenced by Illumina MiSeq (GENEWIZ). Raw sequences for F$_1$ parents (*zpg*–7280) and F$_2$ offspring (*vas2*–5958 and *zpg*–7280) can be found under accession number PRJNA1043640. F$_2$ amplicon sequences were aligned to the F$_1$ parent gene drive sequence in the *zpg*–7280 lines, and to the most common sequence in the *vas2*–5958 lines, using CRISPResso2 (v. 2.2.14)[42]. No restrictions were placed on read quality, in order to preserve the true read proportions for subsequent quality control as detailed below. Complete sequences of all reads found at >0.5% relative abundance be found in Supplementary Data 1.

## Short and long sequencing controls and haplotype calling

Control samples, consisting of six *vas2*–5958/N'Gousso hybrid females, were sequenced at the *zpg*–7280 target site both individually and in a pool to help determine the best method for filtering out erroneous haplotypes from pooled samples, by comparing the haplotypes present in the pooled samples to the known true haplotypes in each pool. These data were used to help determine an appropriate cut off point for distinguishing true haplotypes in the main experimental samples; full methods and results for these controls can be found in Supplementary Material 1.

For the main experiment, reads present at >0.5% relative abundance in each F$_2$ pool were aligned in Benchling[43] to the F$_1$ gene drive and wild type parent chromosomes for *zpg*–7280 lines using MAFFT v7[44]. The most common read in each *vas2*–5958 pool was assumed to match the donor chromosome of the gene drive parent, as was always the case in the *zpg*–7280 samples, and all reads above 0.5% relative abundance were aligned to it also using MAFFT v7. For *vas2*–5958 pools the overall composition of the alleles was used to infer the likely alternate (recipient) chromosome, as allele pools are combinations of chimeric sequences of the true haplotypes. For more detail see Supplementary Information 2 and Supplementary Figs. 5–10.

Haplotype origins in the offspring pools were determined using the presence or absence of SNPs between the parental donor (G3) and recipient (N'Gousso) sequences (Figs. 1, 2). Sequences between a present and an absent SNP were classed as ambiguous. Sequences up to the first SNP from the cut site were assumed to be derived from the donor chromosome if the first SNP matched the donor chromosome

## Table 1 | Mosquito strains used for crosses

| Wild type strains | | | |
|---|---|---|---|
| **Strain** | **Species** | **Place and date of collection** | **Refs.** |
| G3 | *An. gambiae/An. coluzzii* hybrid | The Gambia, 1975 | 11 |
| N'Gousso | *An. coluzzii* | Yaoundé, Cameroon, 2002 | 34 |
| Gene drive lines | | | |
| **Line** | **Locus/target** | **Strain** | **Cassette** | **Refs.** |
| *vas2*-5958 | AGAP005958 (*yellow-g*) | G3 | *vas2*-Cas9, U6-gRNA, 3xP3-RFP | 11 |
| *zpg*–7280 | AGAP007280 (*nudel*) | G3 | *zpg*-Cas9, U6-gRNA, 3xP3-RFP | 21 |

All gene drive lines were created by and obtained from Hammond et al. (2016 and 2021); the G3 strain was obtained from Imperial College London, and the N'Gousso strain obtained from Liverpool School of Tropical Medicine.

## Table 2 | Details of mosquito crosses to produce hybrid parents and their associated offspring pools

| F$_0$ crosses | F$_1$ crosses | # F$_2$ offspring pools sequenced | # F$_2$ offspring pools analysed |
|---|---|---|---|
| 10 ♂ *vas2*–5958 × 25 ♀ N'Gousso | 40 ♂ *vas2*–5958/N'Gousso F$_1$ × 20 ♀ G3 | 6 | 4 |
| 10 ♂ *zpg*–7280 × 25 ♀ N'Gousso | 20 ♀ *zpg*–7280/N'Gousso F$_1$ × 40 ♂ G3 | 6 | 6 |

sequence, or ambiguous if the first SNP matched the recipient chromosome. Haplotypes matching a true sequence except for one erroneous SNP (likely generated during PCR or sequencing) were grouped together with their evident haplotype of origin, with haplotype relative abundance given as a summation of abundances from these sequences. Alleles present at low frequency which match the parent sequences are, based on the results of the control study, most likely to be artifacts of PCR template switching or error (Supplementary Material 2) but are also indistinguishable from minor homed alleles that may occur in a small fraction of germ cells in the parent. For this reason, they are presented in Figs. 4–6 but excluded from analysis using relative abundances of reads, unless stated. Full methodology for the filtering and resolution of haplotypes can be found in the Supplementary Material 2. Graphs and figures were produced using ggplot2 (v. 3.3.6)[45] in R (v. 4.1.2)[46,47] and BioRender (biorender.com).

In addition, amplicons of 4 kb either side of the cut site were sequenced from all six $zpg$−7280 $F_1$ parents and $F_2$ pool gene drive chromosomes to identify potential long resection events not detectable using the shorter ~400 bp amplicons. Reactions were performed using Q5 Hot Start High Fidelity Master Mix (NEB, Cat. # M0494S), with an initial denaturation step at 98 °C for 30 s, followed by 35 cycles of 98 °C for 10 s, 66 °C for 30 s and 72 °C for 1 min 20 s, and a final elongation step at 72 °C for 10 min. Amplicon bands were purified from gel electrophoresis as described above and sent for Nanopore long read sequencing (Source Bioscience).

Demultiplexed long-read fastq files were run through the EPI2ME wf-amplicons workflow (v. 1.0.2); $F_1$ parent samples were run first using default settings to produce a consensus sequence, which was mapped to the short-read parent sequences in Benchling to confirm that the sequence matched the donor chromosome from the gene drive parent. The $F_1$ parent consensus sequence was then used as the reference sequence to run the wf-amplicons workflow on corresponding $F_2$ pools, with the following parameters differing from default: read downsampling = 3000, minimum read length = 500, minimum average read quality = 12. The produced bam files were used to generate variant allele frequencies using bam-readcount (v 1.0.1)[48] and these were plotted using ggplot2 in R.

## Statistics and reproducibility

No statistical method was used to predetermine sample size; limits were placed based on sequencing constraints. Data was only excluded in the specific cases outlined and justified in the manuscript. The pools taken forward for sequencing were chosen to give a range of gene drive inheritance rates, therefore the experiments were not randomised and the investigators were not blinded to allocation during experiments and outcome assessment. Normality of data was checked using a Shapiro-Wilk test for each of the following analyses, confirming that parametric analyses were appropriate. Difference between observed and expected donor haplotypes in offspring pools was interrogated using a paired t-test. Difference in donor sequence frequencies between the two gene drive sites was tested with an unpaired two-sample t-test. All stats were non-directional and performed in R (v 4.3.1)[46,47].

## Reporting summary

Further information on research design is available in the Nature Portfolio Reporting Summary linked to this article.

## Data availability

All raw sequencing data is publicly available in the NIH Sequence Read Archive under accession number PRJNA1043640. Source data are provided with this paper.

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

## Acknowledgements

The Academy of Medical Sciences supported T.N. and P.P. with a Springboard award (SBF006\1183). G.B. was supported by a Torno Subito fellowship (ID 18848) from the Regione Lazio, Italy. T.N. and J.S. were also supported by the Medical Research Council (MR/W002159/1).

## Author contributions

P.P. designed and conducted the data analysis and wrote the manuscript; G.B. assisted on experimental design and conducted experiments; A. A. conducted experiments; J.S. conducted crosses and assisted with data analysis; R.S. conducted experiments; F.L. supervised work; T.N. conceived the study and its design, supervised work and contributed to the writing of the manuscript. All co-authors edited manuscript.

## Competing interests

The authors declare the following competing interests: Tony Nolan has equity in Biocentis. The remaining authors declare no competing interests.
