## [Peer Review File · Nature Communications]

REVIEWER COMMENTS

Reviewer #1 (Remarks to the Author):

In this study on *Anopheles gambiae* mosquitoes, Pescod et al. examine the molecular process of homing of a transgenic gene drive cassette (GD) into a wild-type chromosome. Two distinct gene drive constructs inserted in two different genomic loci are used for this investigation. Genetic diversity between two *Anopheles* strains is ingeniously exploited to identify stretches of DNA originating from either the transgenic or wild-type parent. The goal was to quantify the extent of genomic sequence flanking the transgene that gets co-transferred into the wild-type chromosome during GD homing, on either side of the transgene (gene conversion tracts). This question is important to shed light on the precise genetic changes that potential future GD interventions could introduce into field mosquito populations, in particular the extent to which linked genomic sequences could hitch-hike with a GD. The results clearly show that about 60% of homing events are accompanied by flanking tracts of converted sequence of less than 50 bp. Virtually all conversion events transfer less than a few hundreds of basepairs of flanking sequence. This new knowledge will facilitate the evaluation and risk assessment of GD technologies, by providing a more precise understanding of the molecular process at play during homing of a GD. To my knowledge this is the first careful molecular analysis of this kind in *Anopheles* mosquitoes. In ref. 25 (Verkuijl et al., Nat Comm 2022), a related study has been conducted in *Aedes aegypti* mosquitoes, with a more limited sample size and focusing only on the transfer of nucleotides from in-vitro produced DNA to the genome during the initial transgenesis steps, whereas here the process examined is genomic transfer from a transgenic to a non-transgenic chromosome in a diploid germ cell. The results establish that in general very short stretches of genomic DNA are co-transferred on either side of the homing transgene, providing a clear answer to a long-standing question in the field.

The conclusions are well-supported by the data. My main criticism is that the text is not always immediately easy to understand and would benefit from some editing work to clarify many sentences, otherwise uninformed readers may have difficulties to grasp some parts. I am attaching a version of the manuscript with some annotations and suggestions to improve clarity.

Specific comments:

- The title itself is a bit cryptic to non-specialists. To make it slightly more readily understandable, how about: *Anopheles* homing-based gene drives convert wild-type chromosomes with limited flanking sequence carryover from source chromosomes? I find the term “introgress” misleading in the title (see below) as introgression is not exactly what has been investigated here.

- Abstract: see some clarification suggestions in the attached pdf file sticky notes. The first sentence of the abstract is an example of a confusing sentence: it is easy to misunderstand that the target site is not on the chromosome opposing the GD. Also, not all gene drives target an essential gene. Suggestion : CRISPR-Cas9 gene drive control strategies use a homing selfish genetic element that induces a double-stranded break, and is copied, into a target site on the opposing chromosome, resulting in super-Mendelian inheritance. The beginning of the last sentence of the abstract could refer more explicitly to homing-based as opposed to meiotic drive processes. Suggestion: we confirm that homing, rather than destruction of target chromosomes, is the predominant drive mechanism, and show...

In Genetics terms I somewhat disagree with the term “introgression” in the last sentence of the abstract and in the title. The idea here is that the GD inserts neatly into a WT background. True introgression, a concept of long-term population dynamics, would take more generations as the GD progresses and source chromosomes are lost. In the context of a GD field release, the fact that laboratory chromosomes will be rapidly diluted to negligible frequency, and the notion of “clean introgression”, could be mentioned in the discussion near line 431.

- Methodology question: might PCR artefacts generate mosaic amplicons, e.g. by template switching during the PCR reaction? If such a phenomenon exists, it would confound the analysis and question the validity of some results. Amplicon sequencing from the F1 parents excluding gonadal tissue would allow to control that this is not the case (it could also be performed on the F2 pool DNA on a control SNP-rich locus unrelated to the GD).

- The text does not call figures 3 and 4.

- In Figure 5E, were there really no 2B-haplotypes on the right side of the insertion? Or is an asterisk missing on top of this bar? In Fig. 5A, 5B and to some extent 5D, I'm surprised that the rate of haplotypes 1A on the right side is so much lower than 50%, since parental donor allele inheritance should be around 50%?

- Table 2: typo, female x female cross

- Line 240: not Sanger

- Methods line 252-259 need some clarification, notably about unexpected SNPs: could SNPs not found in either parent be attributed to PCR and sequencing errors? What was the relative abundance of PCR / sequencing errors?

- Lines 261-262: a possible explanation would be welcome for the inconsistency of SNPs in two larval pools.

- Line 387 : do you mean Type 1 instead of Type 2 ?

- Line 476: maybe indicate that the data obtained here does not show appreciable levels of meiotic drive, but that low rates of meiotic drive cannot be ruled out?

Sincerely,

Eric Marois

Reviewer #2 (Remarks to the Author):

Gene drives are a promising way to modify or suppress pest species. Suppression gene drives in the *Anopheles* mosquito have been particularly effective. This study examines two suppression gene drives in *Anopheles gambiae*, differing in their promoter for Cas9 expression. After gene drive takes place, the authors examine the DNA on either side of the new drive alleles, checking which nucleotides have come together with the gene drive from the original drive allele, and which remained identical to the sequence on the original target chromosome. The authors conclude that some sequences flanking the drive could be transferred together with the drive, but that such sequences would usually be short. Thus, a gene drive will quickly transfer from a laboratory background to the wild-type background where the gene drive is released.

Overall, the study seems to have accomplished its objective without any issue, but I believe that its impact is well short of the desired level for Nature Communications. I base my assessment on the following points.

1. The mechanism of homology-directed repair is well-conserved between species. End resection is almost always a few hundred nucleotides, or in rare instances somewhat longer (perhaps a few thousand in extreme cases), as the authors review in the introduction. Thus, the results here are well within expectations.

2. Near the gene drive, even assuming very long HDR, only represents a tiny fraction of the genome. Thus, whether the result was 66% within 50 bp or 30% within 500 bp, or even an imaginary extreme example of 100% with 4000 bp transfers, the general idea of the gene drive moving to a completely different chromosome without any significant nucleotide transfer would still be >99.99% correct.

3. Genetic background can certainly have large fitness effects, but these suppression gene drives are designed to break the target gene in the first place. Genetic background in the local area of the drive is thus of much less importance (because it could not substantially affect a broken gene). It might be a little more important for modification drives, but because these tend to bring their own rescue sequence with them, you are already introducing a large number of changes (making a few SNPs near the specific target gene relatively less important).

4. The specific suppression drives investigated are not the latest designs that target doublesex, so there was no possibility of finding potentially interesting/unusual target-specific phenomena that might be relevant for dsx-based systems, which represent the best gene drives so far.

5. Two previous studies that the authors use to motivate some of their investigations of other aspects likely are problematic (see below).

Though there are certainly a few points of interest in this manuscript, these are more minor points (such as the type 3 haplotypes proportions), not the main topic. I'd thus recommend the manuscript be published in a journal that avoids decisions based on impact, or perhaps a more focused field-specific journal. As for the details, it's in decent shape. I have the following specific comments.

Abstract: Not all gene drives are designed to break an essential gene in a certain sense. Less good gene drives could just target anywhere, and modification drives would not want to "break" the gene in the sense that the new allele won't have a functional copy. Consider removing this part of specifying that it is for suppression drive.

Line 82: These look like typical inheritance rates, not homing rates.

Line 88 and others: "HEG gene" should just be "HEG".

Line 107-108: It's not technically necessary to restrict activity to the germline to get 100% inheritance.

Line 126: This reference had a questionable result based on data from an older paper that could not be replicated, even with the same gene drive. I'd hesitate to base anything important on this one isolated and very strange data point. It would also be quite extraordinary if a single DNA break made a whole cell non-viable without the ability to conduct proper repair.

Line 167: This reference was about knock-in, not gene drive. It was also a single experiment, so its conclusion should be taken with a large grain of salt considering the high potential variability during the insect microinjection process.

Line 194: The Hammond et al. 2016 paper (reference 11) just made one of the gene drives. The other was in Hammond et al. 2021 (reference 21), correct?

Line 372: This is not quite correct. This study had examples with one mode of inheritance or the other, but not both together.

Figure 3-4: It's a little unclear that these different panels are the same experiment but with different parents. Perhaps "A, B, C..." Could be replaced with "Parent A, Parent B, etc...".

Reviewer #3 (Remarks to the Author):

In this manuscript, authors have undertaken to determine the extent to which gene conversion tracts (GCTs) are associated with CRISPR-induced cuts followed by homology-directed repair in *An. gambiae*. The authors show that most GCTs are short, thus facilitating rapid integration into the local genetic background. It appears that long GCTs (>150bp), which could cause the laboratory genetic background to stay in the natural population, are relatively rare, but the authors cannot be certain that even larger GCTs (>400bp) are not present, due to a lack of resolution in the technology used in this study. The authors also show that a substantial number of cut and repair events are associated with non-canonical repair, potentially involving heteroduplex DNA, also highlighting that our understanding of DNA repair mechanisms, particularly in *An. gambiae*, is lacking, and requires further study.

This work, to the best of my knowledge, presents the first systematic investigation of the mechanism of integration of gene drive elements from the donor chromosome into the recipient chromosome in *Anopheles*, and is therefore of significance to the field.

The conclusions made by the authors are largely substantiated by the work. At the same time, it would have been nice for the authors to perform additional experiments in order to gain resolution on some observations within the scope of the study. For example, additional evidence could be gained to resolve the observations of non-canonical integration events, and to differentiate between the potential reasons for the greater-than-expected frequencies of unaltered G3 chromosomes in the F2 progeny (ie. whether this is caused by meiotic drive, or potentially caused by a disproportionate appearance of very long GCTs).

The authors have assumed that the parental genotype is the “most frequent haplotype” among the F2 progeny. Figure 5 prompts the question of whether it is reasonable to assume this. In the zpg-7280 cross, in fact the most common haplotype on the Right side of the gene drive element is not the parental haplotype, but is instead the gene drive element with a small GCT. Is it possible that small GCTs are thus under-represented in the vas2-5958 crosses because the author has biased their findings against seeing these as the most dominant haplotype? Indeed, small GCTs are far less frequent in the F2 progeny of these vas2-5958 crosses.

The authors are also limited in their ability to differentiate between certain possibilities that are important to make their claims. Although it is true that observations of GCTs >150bp (and <400bp) are not common, they reason that because of this, large GCTs are unlikely to be significantly represented in the population. Probably this is a reasonable assumption, but this study also highlights the surprising nature of DNA repair by demonstrating the frequency on non-canonical repair events, and it may thus be worth taking caution in this assumption. In some F2 pools, the Type 1A observations are disproportionately common, and given that an accurate description of the frequency of long GCTs is key to the authors’ conclusions, it would be ideal if they could sequence a longer region of the genome in at least one pool of progeny (ex in Figure 5E) to determine if any of these individuals do have long GCTs, and the proportion of the population that they represent.

Additionally, but less importantly, the authors suggest that some non-canonical events are caused by non-continuous heteroduplex DNA resolution. Given that these events occur with relatively high frequency (7.4% of progeny), would it be possible to visualize these events using fluorescent microscopy to support these conclusions and learn more about these events?

Minor/other comments:

The introduction is very clear and well-written!

Line 206: add the word 'in' "for this reason in zpg-7280..."

Line 255: How frequent were these haplotypes? And these are discarded because they are presumed to be sequencing errors?

Table 3: For the Type 3 GCT events, is it not accurate to give a range of the GCT size ex 151-400bp?

REVIEWER COMMENTS

We would like to thank all three reviewers for their positive appraisal of our work and, in particular, for their excellent comments and critiques; their suggestions for additional work have greatly improved the manuscript and helped refine the analysis of the work as well as the clarity of its exposition. We will address each comment in blue text.

Reviewer #1 (Remarks to the Author):

In this study on *Anopheles gambiae* mosquitoes, Pescod et al. examine the molecular process of homing of a transgenic gene drive cassette (GD) into a wild-type chromosome. Two distinct gene drive constructs inserted in two different genomic loci are used for this investigation. Genetic diversity between two *Anopheles* strains is ingeniously exploited to identify stretches of DNA originating from either the transgenic or wild-type parent. The goal was to quantify the extent of genomic sequence flanking the transgene that gets co-transferred into the wild-type chromosome during GD homing, on either side of the transgene (gene conversion tracts). This question is important to shed light on the precise genetic changes that potential future GD interventions could introduce into field mosquito populations, in particular the extent to which linked genomic sequences could hitch-hike with a GD. The results clearly show that about 60% of homing events are accompanied by flanking tracts of converted sequence of less than 50 bp. Virtually all conversion events transfer less than a few hundreds of basepairs of flanking sequence. This new knowledge will facilitate the evaluation and risk assessment of GD technologies, by providing a more precise understanding of the molecular process at play during homing of a GD. To my knowledge this is the first careful molecular analysis of this kind in *Anopheles* mosquitoes. In ref. 25 (Verkuijl et al., Nat Comm 2022), a related study has been conducted in *Aedes aegypti* mosquitoes, with a more limited sample size and focusing only on the transfer of nucleotides from in-vitro produced DNA to the genome during the initial transgenesis steps, whereas here the process examined is genomic transfer from a transgenic to a non-transgenic chromosome in a diploid germ cell. The results establish that in general very short stretches of genomic DNA are co-transferred on either side of the homing transgene, providing a clear answer to a long-standing question in the field.

The conclusions are well-supported by the data. My main criticism is that the text is not always immediately easy to understand and would benefit from some editing work to clarify many sentences, otherwise uninformed readers may have difficulties to grasp some parts. I am attaching a version of the manuscript with some annotations and suggestions to improve clarity.

We would like to thank Dr Marois for his insightful comments and detailed review of the manuscript, they greatly helped the clarity and structure of the writing as well as helping refine the work itself.

Specific comments:

- The title itself is a bit cryptic to non-specialists. To make it slightly more readily understandable, how about: *Anopheles* homing-based gene drives convert wild-type chromosomes with limited flanking sequence carryover from source chromosomes? I find the term “introgress” misleading in the title (see below) as introgression is not exactly what has been investigated here.

We have discussed our use of the term ‘introgression’ throughout the manuscript, and while it’s true that this is not classical introgression of an allele through repeated backcrossing and natural transfer, the effect of transfer of alleles from an introduced strain into another population could be similar albeit over a much shorter timescale. That being said, we have changed it to ‘transfer’ throughout to avoid conflating the two. We have also clarified the title to be more understandable to non-

specialists.

- Abstract: see some clarification suggestions in the attached pdf file sticky notes. The first sentence of the abstract is an example of a confusing sentence: it is easy to misunderstand that the target site is not on the chromosome opposing the GD. Also, not all gene drives target an essential gene. Suggestion : CRISPR-Cas9 gene drive control strategies use a homing selfish genetic element that induces a double-stranded break, and is copied, into a target site on the opposing chromosome, resulting in super-Mendelian inheritance. The beginning of the last sentence of the abstract could refer more explicitly to homing-based as opposed to meiotic drive processes. Suggestion: we confirm that homing, rather than destruction of target chromosomes, is the predominant drive mechanism, and show...

The comments on the main article have been addressed in the text, with clarifications made where necessary. The abstract has also been changed to better reflect the results and more clearly state the inheritance bias mechanisms.

In Genetics terms I somewhat disagree with the term “introgression” in the last sentence of the abstract and in the title. The idea here is that the GD inserts neatly into a WT background. True introgression, a concept of long-term population dynamics, would take more generations as the GD progresses and source chromosomes are lost. In the context of a GD field release, the fact that laboratory chromosomes will be rapidly diluted to negligible frequency, and the notion of "clean introgression", could be mentioned in the discussion near line 431.

See above for our comments on the use of the term introgression – we agree that as this is not true introgression on a chromosomal level the term may not be appropriate in this context.

- Methodology question: might PCR artefacts generate mosaic amplicons, e.g. by template switching during the PCR reaction? If such a phenomenon exists, it would confound the analysis and question the validity of some results. Amplicon sequencing from the F1 parents excluding gonadal tissue would allow to control that this is not the case (it could also be performed on the F2 pool DNA on a control SNP-rich locus unrelated to the GD).

This was an excellent suggestion for an appropriate control experiment; the methods and thresholds for differentiating between real and artificial alleles in the amplicon sequencing were something we had struggled with in the experimental design. We have performed amplicon sequencing on six hybrid females individually and as a DNA pool to help determine appropriate ways to filter out erroneous sequences in the main samples. This gave us an excellent insight into the structure and composition of our samples, and allowed more stringent and accurate allele identification. While the message of the results (low rates of genetic transfer of gene drive-adjacent sequences from donor to recipient) remains the same, the more complex genetic transfer types were identified as likely to be mosaic artifacts produced during PCR – in depth analysis of the structure of these control pools can be found in the supplementary material. This also allowed us to greatly simplify the description of alleles produced by homing, making the manuscript clearer in general.

- The text does not call figures 3 and 4.

This has been corrected.

- In Figure 5E, were there really no 2B-haplotypes on the right side of the insertion? Or is an asterisk missing on top of this bar? In Fig. 5A, 5B and to some extent 5D, I'm surprised that the rate of haplotypes 1A on the right side is so much lower than 50%, since parental donor allele inheritance

should be around 50%?

The lack of 2B haplotypes on the right side should make more sense now that the analysis has been redone to remove erroneous alleles.

With regards to the unexpectedly low rate of 1A (donor) haplotypes on the right hand side of three of the sample pools (A, B and D), long-read sequencing suggested to confirm the absence of longer resections showed that these three pools contained a previously unknown SNP in the primer region on the donor sequences, explaining the underrepresentation of donor sequences in these pools. Pools A, B and D were therefore excluded from any analysis of the relative proportions donor and recipient chromosomes inherited by offspring.

- Table 2: typo, female x female cross

Corrected

- Line 240: not Sanger

It was actually Sanger sequencing that was used for the F_1 hybrids, as these were individual samples and only one sequence was expected from each amplicon it was possible to use Sanger for these. The pooled F_2 samples were sequenced with Illumina sequencing.

- Methods line 252-259 need some clarification, notably about unexpected SNPs: could SNPs not found in either parent be attributed to PCR and sequencing errors? What was the relative abundance of PCR / sequencing errors?

With the improved filtering and screening procedure based on the new controls this was no longer an issue. An improved detailed description of the screening process for alleles from pooled samples has been given in the supplementary material, for maximum transparency.

- Lines 261-262: a possible explanation would be welcome for the inconsistency of SNPs in two larval pools.

Some of the *vas2-5958* pools had lower numbers of SNPs between the parent chromosomes than others, with only three or four SNPs in each ~350 bp sequence common. In these two pools it is considered that only one SNP existed between the parent chromosomes in these regions, making it difficult to differentiate any homed alleles in the offspring pools. A more detailed explanation of the process behind this exclusion is given in the supplementary material.

- Line 387 : do you mean Type 1 instead of Type 2 ?

Yes, this was an error – but the analysis based on Type has now been removed to simplify the structure of the text.

- Line 476: maybe indicate that the data obtained here does not show appreciable levels of meiotic drive, but that low rates of meiotic drive cannot be ruled out?

Stronger evidence for some contribution from meiotic drive to the biased inheritance rate has now been added to the paper, in the form of long-read sequencing. This has been detailed in the text, and the discussion has been changed to reflect this.

Sincerely,
Eric Marois

Reviewer #2 (Remarks to the Author):

Gene drives are a promising way to modify or suppress pest species. Suppression gene drives in the *Anopheles* mosquito have been particularly effective. This study examines two suppression gene drives in *Anopheles gambiae*, differing in their promoter for Cas9 expression. After gene drive takes place, the authors examine the DNA on either side of the new drive alleles, checking which nucleotides have come together with the gene drive from the original drive allele, and which remained identical to the sequence on the original target chromosome. The authors conclude that some sequences flanking the drive could be transferred together with the drive, but that such sequences would usually be short. Thus, a gene drive will quickly transfer from a laboratory background to the wild-type background where the gene drive is released.

Overall, the study seems to have accomplished its objective without any issue, but I believe that its impact is well short of the desired level for Nature Communications. I base my assessment on the following points.

1. The mechanism of homology-directed repair is well-conserved between species. End resection is almost always a few hundred nucleotides, or in rare instances somewhat longer (perhaps a few thousand in extreme cases), as the authors review in the introduction. Thus, the results here are well within expectations.

We absolutely agree that the broad mechanism for HDR is conserved between species and that the expected outcomes of gene conversion were small in comparison to the wider genome. However, we would add that there are documented differences in the outcomes of HDR between species, even within Diptera (as stated in the introduction). Gene drives have enormous potential for vector control at scale and will therefore be rightfully subject to the scrutiny for efficacy and predictability given to all control strategies used on wild populations (e.g. insecticides, which despite long term use are still under continuous examination). Knowledge and documentation of the precise outcomes of gene drive homing, even within a few hundred or thousand bases, will be essential for regulators to be assured of the predictability of widespread gene drive use regardless of the phenotypic impacts.

There are also positive use cases for this knowledge in terms of gene drive design. A wider range of targets can be used with the knowledge that important SNPs more than ~150 bp away will almost certainly not be forced into homozygosity in a wild population, and gene drives may be designed to intentionally force immediately adjacent SNPs into homozygosity. While such designs were plausible based on knowledge of DNA repair before this study, we believe this work provides much needed evidence and quantification of the outcomes of gene drive repair that will allow the above concepts to be designed effectively.

2. Near the gene drive, even assuming very long HDR, only represents a tiny fraction of the genome. Thus, whether the result was 66% within 50 bp or 30% within 500 bp, or even an imaginary extreme example of 100% with 4000 bp transfers, the general idea of the gene drive moving to a completely different chromosome without any significant nucleotide transfer would still be >99.99% correct.

See above

3. Genetic background can certainly have large fitness effects, but these suppression gene drives are designed to break the target gene in the first place. Genetic background in the local area of the drive is thus of much less importance (because it could not substantially affect a broken gene). It might be a little more important for modification drives, but because these tend to bring their own rescue sequence with them, you are already introducing a large number of changes (making a few SNPs near the specific target gene relatively less important).

While it is true that a novel SNP introduced into a broken gene will be unlikely to have a phenotypic effect, it is still important to be able to predict repair outcomes on the large scale at which they would happen if gene drives were used in control strategies. Documented, peer-reviewed knowledge of the outcomes of gene drive homing is essential for their regulation. There are also several gene drives in development (in our lab and others) which involve modification of target sites rather than breaking of target genes; predictability of these repair outcomes is essential for the function of modification-based gene drives. This is particularly true for the conserved genes that all modern gene drives target, where single SNPs can have profound phenotypic effects (e.g. insecticide target site modifications which confer resistance).

4. The specific suppression drives investigated are not the latest designs that target doublesex, so there was no possibility of finding potentially interesting/unusual target-specific phenomena that might be relevant for *dsx*-based systems, which represent the best gene drives so far.

Respectfully, we disagree with the premise that there would be nothing interesting or relevant to other gene drive systems in this work. We have shown that two different gene drive targets with different germline Cas9 promoters will have broadly similar outcomes, in terms of small conversion tract frequency and potential for combined homing and meiotic drive. This suggests that these results may be applicable to other Cas9-based gene drives in *Anopheles*, which would include the latest *doublesex*-targeting designs.

5. Two previous studies that the authors use to motivate some of their investigations of other aspects likely are problematic (see below).

While these papers were not motivating factors for the study, and were discovered after its design, their inclusion in the literature review aspect of the paper has been addressed in turn below.

Though there are certainly a few points of interest in this manuscript, these are more minor points (such as the type 3 haplotypes proportions), not the main topic. I'd thus recommend the manuscript be published in a journal that avoids decisions based on impact, or perhaps a more focused field-specific journal. As for the details, it's in decent shape. I have the following specific comments.

Abstract: Not all gene drives are designed to break an essential gene in a certain sense. Less good gene drives could just target anywhere, and modification drives would not want to "break" the gene in the sense that the new allele won't have a functional copy. Consider removing this part of specifying that it is for suppression drive.

This is true, and has been corrected.

Line 82: These look like typical inheritance rates, not homing rates.

Thank you, this has been corrected

Line 88 and others: "HEG gene" should just be "HEG".

This has been corrected

Line 107-108: It's not technically necessary to restrict activity to the germline to get 100% inheritance.

True, this has been reworded.

Line 126: This reference had a questionable result based on data from an older paper that could not be replicated, even with the same gene drive. I'd hesitate to base anything important on this one isolated and very strange data point. It would also be quite extraordinary if a single DNA break made a whole cell non-viable without the ability to conduct proper repair.

We absolutely agree, this was an unusual paper and the fact that the results from the first experiment were not reproducible calls into question the idea that meiotic drive can selectively occur instead of homing in a Cas9-based gene drive. However, the intention of our study was originally to discover the proportion and limits of gene conversion during homing; the idea of meiotic drive playing a role in a homing-based gene drive was discussed and subsequently investigated once the initial results clearly showed a bias towards the donor chromosome in the offspring pools.

In our study we find no evidence to suggest that meiotic drive causes 100% of the inheritance bias in some individuals and not others. Instead we find evidence to suggest that chromosomes may be removed, at a low frequency, by a Cas9-induced double stranded break. This makes no difference to the number of offspring inheriting the gene drive, as gametes are in abundance and would therefore not limit fecundity, but would only be visible in an experiment like this where the donor and recipient chromosome are tracked. We therefore hope the reviewer can agree that, while we include the above paper to give a complete picture of the current literature, the suggested role of meiotic drive is different.

Line 167: This reference was about knock-in, not gene drive. It was also a single experiment, so it's conclusion should be taken with a large grain of salt considering the high potential variability during the insect microinjection process.

This is true, and the text has been edited to reflect both of these facts.

Line 194: The Hammond et al. 2016 paper (reference 11) just made one of the gene drives. The other was in Hammond et al. 2021 (reference 21), correct?

Thank you, this has been corrected.

Line 372: This is not quite correct. This study had examples with one mode of inheritance or the other, but not both together.

This was poorly worded, and has been changed.

Figure 3-4: It's a little unclear that these different panels are the same experiment but with different parents. Perhaps "A, B, C..." Could be replaced with "Parent A, Parent B, etc...".

Thank you, we have changed the panel labels to “Pool A, etc.” to make this clearer.

Reviewer #3 (Remarks to the Author):

In this manuscript, authors have undertaken to determine the extent to which gene conversion tracts (GCTs) are associated with CRISPR-induced cuts followed by homology-directed repair in *An. gambiae*. The authors show that most GCTs are short, thus facilitating rapid integration into the local genetic background. It appears that long GCTs (>150bp), which could cause the laboratory genetic background to stay in the natural population, are relatively rare, but the authors cannot be certain that even larger GCTs (>400bp) are not present, due to a lack of resolution in the technology used in this study. The authors also show that a substantial number of cut and repair events are associated with non-canonical repair, potentially involving heteroduplex DNA, also highlighting that our understanding of DNA repair mechanisms, particularly in *An. gambiae*, is lacking, and requires further study.

This work, to the best of my knowledge, presents the first systematic investigation of the mechanism of integration of gene drive elements from the donor chromosome into the recipient chromosome in *Anopheles*, and is therefore of significance to the field.

The conclusions made by the authors are largely substantiated by the work. At the same time, it would have been nice for the authors to perform additional experiments in order to gain resolution on some observations within the scope of the study. For example, additional evidence could be gained to resolve the observations of non-canonical integration events, and to differentiate between the potential reasons for the greater-than-expected frequencies of unaltered G3 chromosomes in the F2 progeny (ie. whether this is caused by meiotic drive, or potentially caused by a disproportionate appearance of very long GCTs).

The authors have assumed that the parental genotype is the “most frequent haplotype” among the F2 progeny. Figure 5 prompts the question of whether it is reasonable to assume this. In the zpg-7280 cross, in fact the most common haplotype on the Right side of the gene drive element is not the parental haplotype, but is instead the gene drive element with a small GCT. Is it possible that small GCTs are thus under-represented in the vas2-5958 crosses because the author has biased their findings against seeing these as the most dominant haplotype? Indeed, small GCTs are far less frequent in the F2 progeny of these vas2-5958 crosses.

The authors are also limited in their ability to differentiate between certain possibilities that are important to make their claims. Although it is true that observations of GCTs >150bp (and <400bp) are not common, they reason that because of this, large GCTs are unlikely to be significantly represented in the population. Probably this is a reasonable assumption, but this study also highlights the surprising nature of DNA repair by demonstrating the frequency on non-canonical repair events, and it may thus be worth taking caution in this assumption. In some F2 pools, the Type 1A observations are disproportionately common, and given that an accurate description of the frequency of long GCTs is key to the authors’ conclusions, it would be ideal if they could sequence a longer region of the genome in at least one pool of progeny (ex in Figure 5E) to determine if any of these individuals do have long GCTs, and the proportion of the population that they represent.

Additionally, but less importantly, the authors suggest that some non-canonical events are caused by non-continuous heteroduplex DNA resolution. Given that these events occur with relatively high frequency (7.4% of progeny), would it be possible to visualize these events using fluorescent

microscopy to support these conclusions and learn more about these events?

We would like to thank reviewer 3 for their suggestions of further work, in particular the suggestion to sequence a larger region around the gene drive insert in order to differentiate between long resections and meiotic drive in the sequences matching the donor chromosome. We implemented this suggestion for all six pools of the *zpg-7280/N'*Gousso offspring pools, and in doing so discovered a cryptic SNP in the primer sequence for three pools that explained the unexpectedly low proportion of donor sequences in these pools. Additionally, by sequencing a longer region we were able to provide additional evidence for the contribution of meiotic drive to inheritance bias, by ruling out any substantial contribution of long conversion tracts.

In response to the comment about whether it is unwise to assume the most common allele must be the donor haplotype, we planned to put this suggestion into effect by labelling the *vas2-5958/N'*Gousso haplotypes Origin 1 and Origin 2, rather than assigning them to donor or recipient. However, upon re-analysis of the data in light of the control experiment suggested by reviewer 1, we discovered that the origin of the haplotype was either obvious due to the expected small resections (Pool H, left of the cut site), or that there were only two haplotypes identified in the pool. In these samples there was therefore no potential for misinterpretation or loss of resolution of small GCTs. We hope that reviewer 3 will agree on this point, but would be happy to discuss it further.

Minor/other comments:

The introduction is very clear and well-written!

Thank you!

Line 206: add the word 'in' "for this reason in *zpg-7280*..."

Thank you, corrected

Line 255: How frequent were these haplotypes? And these are discarded because they are presumed to be sequencing errors?

This is a valid point and we have rewritten the supplementary material to be clearer about the process of discarding erroneous alleles.

Table 3: For the Type 3 GCT events, is it not accurate to give a range of the GCT size ex 151-400bp?

Thanks to the control experiments suggested in review these Type 3 events have been identified as PCR artifacts and removed from the experiment – the full process for this is given in the supplementary material.

REVIEWER COMMENTS

Reviewer #1 (Remarks to the Author):

(Word file attached)

As I wrote in my comments to version 1 of this manuscript, I find this study very interesting, cleverly designed, and informative to readers interested in CRISPR and /or gene drive, and to stakeholders of interventions against disease vector mosquitoes. The conclusions of this work will be impactful, particularly the short length of GCTs and the strengthened data pointing to a significant contribution of meiotic drive during gene drive homing, making this work important to share with the research community.

However, while I am happy with the authors' response to my previous comments, I think the manuscript could still mature some more. I found some parts of the revised text hard to follow, some further clarifications would help. Minor edits and providing of a bit more detail, giving consideration to how the text could be understood or misunderstood by readers, would suffice for the most part. My main critical point is in the way the data is analysed in Supplemental figures 3 to 6 and, as a consequence, in figures 3 to 5 : I have a problem with the way the frequencies of the various alleles are calculated (details below).

List of points to address :

- I like the new title, but would replace “local genetic backgrounds” by “a natural genetic background” (because “local genetic backgrounds” looks like transfer of the gene drives to a diversity of local strains was investigated, which is not the case).

- Abstract lines 25-26: for Δ that, or contribute Δ contribution

- Lines 59-63: HEG genes Δ HEG (in agreement with Rev. 2)

- Line 78: promotion Δ expression

- Line 169: please briefly explain here how sequences were obtained and analysed to determine the SNPs between the Ngousso and GD chromosomes

- Line 172: bug in the call for Figure 2

- Line 175: it's a bit awkward to mention what happened to pools I and J before explaining the experimental setups that defined the pools

- Line 188: the genetics of the experimental setup should be briefly explicated here so that the reader doesn't have to jump to Methods, please explain how each pool was formed (progeny of a single F1 hybrid backcrossed to G3)

- Figure 3: for more precision and clarity, the title of the figure should be: Offspring haplotypes from zpg.... F1 hybrid females. Similarly, for Figure 4: Offspring haplotypes from vasa.... F1 hybrid males. In the titles of Suppl. Figures 3-6, it would also be helpful to mention in brackets the GD strain corresponding to these groups of pools (e.g.: Candidate alleles in sample pools J, K and L (vasa..... cross))

- Legend of figure 3: clarify the asterisk sentence to make it more easily understandable (e.g.: ...due to the presence of a SNP on donor chromosomes, in the binding site of the PCR primer used to amplify the genomic sequence)

- In figure 4, pool L: I wonder why there is this big difference in the donor chromosome sequence frequency between the left and right side (0.526 vs. 0.821, even if frequency is adjusted following my suggestions below). Consider indicating a possible explanation.

- Line 225 to maximise clarity, please add: "in the absence of meiotic drive"

- Line 226, equation 1: This equation would be useful if re-used elsewhere in the text or in the Methods, but this doesn't seem to be the case or I missed it. Also, I find the text that introduces this equation confusing: lines 224-226 discuss the proportion of offspring containing the donor haplotype, "this can be expressed as equation 1", but equation 1 gives the proportion of offspring containing a homed copy in the recipient background, i.e., the other type of positives. Finally, in my logic the equation would be easier to grasp if written $H = 1 - \frac{1}{2}(T/P)$, if necessary to keep it. It could

also be moved to Methods and here the text could better explain the analysis leading to Figure 5. Mentioning the long read sequencing bug on lines 233-237 is confusing, as the reader is led to think that figure 5 is based on long read sequencing results.

- Line 240: delete “pools” (it looks like each pool had exactly 74.2% instead of the global average across pools)

- Line 241: “we saw two different haplotypes” but figure 5 seems to show 3 different haplotypes (in pool H)

- Line 249: explicit the statistical test and what is V

- Lines 259-263 and figure 6 : hard to understand

- Figure 6 pool F: is there an explanation why variant allele frequency differs between the right and left sides of the GD? They should be similar if reflecting the presence of higher than expected proportions of donor chromosomes.

- Suppl. Information 1 after Fig S2: what threshold value was finally chosen and how was it chosen, to distinguish real from artefactual alleles in the F2? (but see below)

- The zpg-7280/N’Gousso hybrid F2 offspring sequences were aligned to the parent Ngousso sequence, whereas the vas2-5958/N’Gousso hybrid F2 offspring sequences were mapped to the most common allele in the pool, due to unavailability of the Ngousso parental sequence. This could be explained and justified more explicitly in the Methods and Suppl. Information in addition to the legend of figure 4.

- it’s great that the authors performed the controls shown in Suppl Fig 2, revealing an interference of PCR chimeras in the analysis. Suppl Fig 3-6 are very useful, however, I don’t fully agree with the way the data was categorised and analysed:

all “artefact” alleles showing a chimera between the two parental sequences starting with the donor, ending with the recipient, should be more properly termed “Artefact or minor homed allele”, since it’s impossible to distinguish a PCR chimera from a true minor allele that arose by homing. An

arbitrary frequency threshold based on the result in Suppl Fig 2 to classify an allele as an artefact may exclude real minor alleles formed by homing in only a small fraction of germ cells in the parent, that contributed a minor frequency in the next generation of mosquitoes. Minor homed alleles could also be observed following mating of the parent female (zpg cross) with multiple males, with minor sperm contributors, as can happen in cage conditions.

Some other alleles marked as artefacts evidently belong to one of the major allele classes, with just an extra PCR mutation: in my opinion they should be counted with their corresponding major allele (placing the two alleles next to each other in the figure, with each observed proportion, + indicate the sum of their proportions with a bracket), this would provide more accurate proportions that should be adjusted throughout the manuscript (notably figures 3, 4, 5).

Sequences that are 100% like the recipient allele should not be classified as artefacts, in spite of their low frequency (in pool D left and right, J right), illustrating that an arbitrary frequency threshold is inappropriate to classify a sequence as an artefact.

The only sequences that can most certainly be tagged as PCR chimera artefacts, and firmly excluded from the frequency calculations, are those low-frequency ones that begin with recipient sequence identity, but end with donor sequence identity, which cannot reasonably be the product of homing.

As a result of the uncertainty on what is a PCR chimera or a true allele, the frequencies as shown in figures 3, 4, 5 cannot exactly reflect reality. I think it would be more correct to show an additional category “artefacts or minor homed alleles” in these figures.

- Suppl. Fig. 6: what are the white lines on some of the grey bars in pools G, H, K, L?

- in the discussion near lines 325-328 it would be worth mentioning that the vasa and zpg experiments differ in that vasa investigated gene drive inheritance from males, zpg from females. That results were similar suggests identical homing mechanisms in both sexes.

- Lines 298-302: this seems contradictory with the results, which show that carrying over an adjacent allele, even if desired, cannot happen efficiently.

- Line 309: not clear from the text if reference (25) was irreproducible, or the other reference therein. The older reference should be cited for clarification.

Reviewer #2 (Remarks to the Author):

I am reviewer 2 from the previous round of reviews. The authors have made some useful improvements to the manuscript and fixed some important errors (the suggestion from reviewer 1 and fortunate discovery of the interfering SNP were the most important for this). I'm still unconvinced by the level of advance the paper provides (though see below for one possible exception), but I am willing to certify that it meets scientific soundness for the experiments and writing.

Previous review point 1: if the authors think that the knowledge provided in the paper opens up some avenue of approach or allows greater confidence for some sort of biosafety, I'd encourage them to describe a specific scenario (or scenarios) in the introduction and/or discussion in which more accurate knowledge of tract conversion length is critical to some sort of design or biosafety/regulatory aspect. Right now, I can only think of allelic drives designed to convert a tract with many mutations from a more limited number of gRNA target sites, and these seem to be more specific for pesticide resistance reduction in agricultural pests. In this case, the species-specific knowledge provided here may still not be specific enough because I'm not sure if such drives are being considered for *Anopheles* mosquitoes.

Previous review point 2: I think I may not have been specific enough here. The introduction states that "A common concern is whether the lab-bred mosquito strains containing the gene drive will be capable of rapid transfer into wild populations due to lack of fitness after years of lab maintenance, slowing the initial spread of the gene drive element into the target population." I was saying that the specific rates measured here aren't related to this question because even with the wider uncertainty range before this study narrow it down, it was still small enough for this to be only a tiny fraction of the genome and very unlikely to produce any fitness effects at all, much less significant effects.

Previous review point 3: Again, I'm having trouble seeing specific examples here, aside from the one specific case of allelic drive. As an aside, this specific case should probably be mentioned to try and increase the study's impact, though there may be some differences between conversion tracts when copying a large drive alleles versus just a small mutation.

The response says that, “While it is true that a novel SNP introduced into a broken gene will be unlikely to have a phenotypic effect, it is still important to be able to predict repair outcomes on the large scale at which they would happen if gene drives were used in control strategies.”

I am having trouble seeing the importance of this. Maybe explain with an example?

The response says that, “predictability of these repair outcomes is essential for the function of modification-based gene drives. This is particularly true for the conserved genes that all modern gene drives target, where single SNPs can have profound phenotypic effects.”

Maybe we are still discussing insecticide targeting drives, but the response here seems to indicate that it could be important for other drives. I don’t think that all or even most rescue type modification drives that are essential for viability will have alleles that really need close tracking for this, even if they are the part of the gene or promoter region that’s actually important. Again, the authors may be thinking of an example that I am not imagining that could strengthen their manuscript if it were included.

For your response to my comment in line 126 and my new read of the paper, things seem to be getting more interesting. Apologies if I missed this before (though maybe it’s just apparent in the new data for Figure 6), but this certainly seems to point to uneven inheritance of the two chromosomes. Meiotic drive is invoked as an explanation, which may be accurate. However, I think that a little bit more data is needed to support this, especially because the different batches seem to have different rates of donor chromosome inheritance. It seems to me that in addition to attributing the difference to drive effects, there could be fitness differences between the chromosomes at sperm or offspring stages or even a natural meiotic drive. Unless I misunderstand the situation, it may be necessary to repeat the experiment (for Figure 6) with non-drive mosquitoes of the same hybrid strain and cross in order to be able to firmly attribute the difference to drive effects. If the drive does indeed turn out to be the cause, then it would be very interesting.

One minor point I saw in the new version:

Line 172: “Error! Reference source not found.”

Because my opinion on this study’s impact seemed to be substantially different from the other reviewers, I found a junior colleague to provide a short review, which follows:

In this study, the authors investigated gene conversion tracts (GCTs) in Anopheles gene drive, demonstrating that around two-thirds of GCTs lengths were less than 50 base pairs while the rest was only a few hundred base pairs. They also observed the contribution of meiotic drive to biasing drive inheritance in certain tested pools. The revisions made in response to the previous round of review have improved the manuscript significantly. Overall, the manuscript is well-constructed, with reasonably designed experiments, solid data and clear writing. I only have a few concerns:

To enhance the scientific impact of this study, it may be beneficial to further elucidate its broader application. In addition to the allelic drive mentioned by Reviewer 2, the findings of this study could potentially inform the design of toxin-antidote modification drives (<https://doi.org/10.1038/s41467-020-14960-3>). In such drives, modified SNPs in target sites are intended to be introduced into the genome via copying from homology arms; this may be needed to prevent cleavage of the drive allele. However, if these modifications are situated more than a few hundred base pairs away, their transfer into the genome of *Anopheles* mosquitoes may prove challenging. More examples of potential application may also be discussed to strengthen the impact.

Both short-read and long-read experiments supported the conclusion that meiotic drive existed in certain tested pools. While it is understood that multiple cuts in a chromosome (e.g., *W*-shredders) can disrupt the entire chromosome and induce meiotic drive, the mechanism by which a single cut achieves this is less clear. Providing additional explanations or references for this would enhance the comprehensibility of the manuscript.

Typos:

Line 116: “...the length of donor sequence copied into the recipient...” should be “...the length of donor sequence copied into the recipient...”.

Line 172: “Error! Reference source not found”.

Line 352: Imperial University?

Reviewer #3 (Remarks to the Author):

In this manuscript, authors have undertaken to determine the extent to which gene conversion tracts (GCTs) are associated with CRISPR-induced cuts followed by homology-directed repair in *An. gambiae*. The authors show that most GCTs are short, thus facilitating rapid integration into the local genetic background.

The authors have now undertaken additional experiments to demonstrate more clearly that 1) long GCTs are not going undetected in their analyses and that 2) meiotic drive is the most likely explanation for disproportionate inheritance of the donor (drive) chromosome. These additional studies have significantly improved the manuscript and hone in on greater accuracy in the results.

This work, to the best of my knowledge, presents the first systematic investigation of the mechanism of integration of gene drive elements from the donor chromosome into the recipient chromosome in Anopheles, and is therefore of significance to the field.

Minor comments:

line 62: HEG gene

Line 174: fix reference error

Fig 4 legend: is there a better way to display the inaccurate proportion data? Is it possible to show a range or add a brief description of the degree of inaccuracy?

REVIEWER COMMENTS

We would like to again thank all three reviewers for their constructive critique of this manuscript; the suggestions have been used to improve the readability and accuracy of the work.

To summarise the reviewer comments: Reviewers 1 and 3 are positive about the work and its impact. Reviewer 2 is happy with the conclusions of the work and the soundness of its execution but had some concerns about impact, for which they asked a second opinion of another lab member, who gave a positive assessment. So, all four of the reviewers are confidently positive about the execution of the work, and three of the four are very positive about its impact. We hope therefore that following the latest round of revisions the article will be acceptable for publication in your journal.

All reviewer points have been addressed below in blue text – note, line numbers refer to the document submitted with no tracked changes.

Reviewer #1 (Remarks to the Author):

As I wrote in my comments to version 1 of this manuscript, I find this study very interesting, cleverly designed, and informative to readers interested in CRISPR and /or gene drive, and to stakeholders of interventions against disease vector mosquitoes. The conclusions of this work will be impactful, particularly the short length of GCTs and the strengthened data pointing to a significant contribution of meiotic drive during gene drive homing, making this work important to share with the research community.

However, while I am happy with the authors' response to my previous comments, I think the manuscript could still mature some more. I found some parts of the revised text hard to follow, some further clarifications would help. Minor edits and providing of a bit more detail, giving consideration to how the text could be understood or misunderstood by readers, would suffice for the most part. My main critical point is in the way the data is analysed in Supplemental figures 3 to 6 and, as a consequence, in figures 3 to 5 : I have a problem with the way the frequencies of the various alleles are calculated (details below).

We would like to thank Prof Marois for providing an incredibly thorough and constructive response to this manuscript. We are gratified that he finds the work both of high quality and high impact, and are grateful for the suggestions he has made for adjusting the attribution of read proportions in particular.

List of points to address :

- I like the new title, but would replace “local genetic backgrounds” by “a natural genetic background” (because “local genetic backgrounds” looks like transfer of the gene drives to a diversity of local strains was investigated, which is not the case).

Agreed - changed

- Abstract lines 25-26: for Δ that, or contribute Δ contribution

Corrected

- Lines 59-63: HEG genes Δ HEG (in agreement with Rev. 2)

Corrected

- Line 78: promotion Δ expression

Corrected

- Line 169: please briefly explain here how sequences were obtained and analysed to determine the SNPs between the Ngousso and GD chromosomes

Added

- Line 172: bug in the call for Figure 2

Thank you, this has been fixed.

- Line 175: it’s a bit awkward to mention what happened to pools I and J before explaining the experimental setups that defined the pools

We have added an additional panel to Figure 1 to better describe the pool set up, and outlined the experimental set up in lines 162-170.

- Line 188: the genetics of the experimental setup should be briefly explicated here so that the reader doesn’t have to jump to Methods, please explain how each pool was formed (progeny of a single F1 hybrid backcrossed to G3)

See above comment.

- Figure 3: for more precision and clarity, the title of the figure should be: Offspring haplotypes from zpg.... F1 hybrid females. Similarly, for Figure 4: Offspring haplotypes from vasa.... F1 hybrid males. In the titles of Suppl. Figures 3-6, it would also be helpful to

mention in brackets the GD strain corresponding to these groups of pools (e.g.: Candidate alleles in sample pools J, K and L (vasa..... cross))

Corrected in all figures (Figs 3-5 and Supp Figs 5-10)

- Legend of figure 3: clarify the asterisk sentence to make it more easily understandable (e.g.: ...due to the presence of a SNP on donor chromosomes, in the binding site of the PCR primer used to amplify the genomic sequence)

Corrected

- In figure 4, pool L: I wonder why there is this big difference in the donor chromosome sequence frequency between the left and right side (0.526 vs. 0.821, even if frequency is adjusted following my suggestions below). Consider indicating a possible explanation.

We have attributed this difference between the abundance of donor sequence between left and right of the cut site (which is present in Pool L and also in the long read analysis of Pool F, Figure 7) to a disparity in primer binding, and discussed it in Lines 273-280.

- Line 225 to maximise clarity, please add: “in the absence of meiotic drive”

Added

- Line 226, equation 1: This equation would be useful if re-used elsewhere in the text or in the Methods, but this doesn't seem to be the case or I missed it. Also, I find the text that introduces this equation confusing: lines 224-226 discuss the proportion of offspring containing the donor haplotype, “this can be expressed as equation 1”, but equation 1 gives the proportion of offspring containing a homed copy in the recipient background, i.e., the other type of positives. Finally, in my logic the equation would be easier to grasp if written $H = 1 - \frac{1}{2}(T/P)$, if necessary to keep it. It could also be moved to Methods and here the text could better explain the analysis leading to Figure 5. Mentioning the long read sequencing bug on lines 233-237 is confusing, as the reader is led to think that figure 5 is based on long read sequencing results.

After rereading the section we decided the equation wasn't necessary to convey the concept and doesn't provide any extra clarity, so it has been removed

- Line 240: delete “pools” (it looks like each pool had exactly 74.2% instead of the global average across pools)

Corrected

- Line 241: “we saw two different haplotypes” but figure 5 seems to show 3 different haplotypes (in pool H)

Corrected

- Line 249: explicit the statistical test and what is V

With the alteration of the data (reclassifying artefacts with an obvious origin) a normal distribution was restored, allowing the use of a t test instead of a Wilcoxon non-parametric test (Lines 268 and 282)

- Lines 259-263 and figure 6 : hard to understand

Both reworded for clarity (now Figure 7 and line 294-311)

- Figure 6 pool F: is there an explanation why variant allele frequency differs between the right and left sides of the GD? They should be similar if reflecting the presence of higher than expected proportions of donor chromosomes.

Potential explanations for the difference between donor abundance between left and right in Pools F have been added to the main text (lines 307-311).

- Suppl. Information 1 after Fig S2: what threshold value was finally chosen and how was it chosen, to distinguish real from artefactual alleles in the F2? (but see below)

This has now been explained more fully in Supplementary Information 2

- The zpg-7280/N'Gousso hybrid F2 offspring sequences were aligned to the parent N'Gousso sequence, whereas the vas2-5958/N'Gousso hybrid F2 offspring sequences were mapped to the most common allele in the pool, due to unavailability of the N'Gousso parental sequence. This could be explained and justified more explicitly in the Methods and Suppl. Information in addition to the legend of figure 4.

This has been done in what is now Figure 1, Figure 5 and Lines 474-481 in the main text, and Supplementary Information 2.

- it's great that the authors performed the controls shown in Suppl Fig 2, revealing an interference of PCR chimeras in the analysis. Suppl Fig 3-6 are very useful, however, I don't fully agree with the way the data was categorised and analysed:

all "artefact" alleles showing a chimera between the two parental sequences starting with the donor, ending with the recipient, should be more properly termed "Artefact or minor homed allele", since it's impossible to distinguish a PCR chimera from a true minor allele that arose by homing. An arbitrary frequency threshold based on the result in Suppl Fig 2 to classify an allele as an artefact may exclude real minor alleles formed by homing in only a small fraction of germ cells in the parent, that contributed a minor frequency in the next generation of mosquitoes. Minor homed alleles could also be observed following mating of

the parent female (zpg cross) with multiple males, with minor sperm contributors, as can happen in cage conditions.

Just to note here, minor alleles would not be observed after mating of the *zpg*-7280 female F1 parent to multiple males, as all alleles sequenced are from the female parent not the male parent (by placing one primer within the gene drive element). Minor alleles from multiple male matings could, however, have occurred in *vas2*-5958 offspring pools, where multiple gene drive males may have mated with a single wild type female. In this case four different haplotypes (two donors, two recipients) would have been observed in a single pool; this didn't appear to be the case in any of the six *vas2*-5958 pools we sequenced.

Some other alleles marked as artefacts evidently belong to one of the major allele classes, with just an extra PCR mutation: in my opinion they should be counted with their corresponding major allele (placing the two alleles next to each other in the figure, with each observed proportion, + indicate the sum of their proportions with a bracket), this would provide more accurate proportions that should be adjusted throughout the manuscript (notably figures 3, 4, 5).

Sequences that are 100% like the recipient allele should not be classified as artefacts, in spite of their low frequency (in pool D left and right, J right), illustrating that an arbitrary frequency threshold is inappropriate to classify a sequence as an artefact.

The only sequences that can most certainly be tagged as PCR chimera artefacts, and firmly excluded from the frequency calculations, are those low-frequency ones that begin with recipient sequence identity, but end with donor sequence identity, which cannot reasonably be the product of homing.

As a result of the uncertainty on what is a PCR chimera or a true allele, the frequencies as shown in figures 3, 4, 5 cannot exactly reflect reality. I think it would be more correct to show an additional category "artefacts or minor homed alleles" in these figures.

We would like to thank Prof Marois for this excellent suggestion for the improvement of the classification of reads from the pooled samples to avoid entirely removing potential minor alleles, which would have been excluded with our previous threshold. We have made most of the suggested changes. Any alleles which were identical to one of the known haplotypes except for a single base change were grouped in with their apparent source sequence for the analyses. We then set a threshold for reads which we termed 'minor' alleles – aka reads which were plausible homing events, but also potential PCR chimeras – of <10% of the relative abundance of reads, calculated after removing any reads present at <0.5% of the sample. This threshold was chosen based on examination of the control read

proportions. Minor alleles have been presented in Figures 3-5, differentiated from major alleles by size, and were included in the proportions shown in Figure 6.

- Suppl. Fig. 6: what are the white lines on some of the grey bars in pools G, H, K, L?

White lines in the reference sequence indicate that the SNP at that location is an insertion relative to the reference – this has been added to the legend for Fig S6.

- in the discussion near lines 325-328 it would be worth mentioning that the vasa and zpg experiments differ in that vasa investigated gene drive inheritance from males, zpg from females. That results were similar suggests identical homing mechanisms in both sexes.

Added

- Lines 298-302: this seems contradictory with the results, which show that carrying over an adjacent allele, even if desired, cannot happen efficiently.

Clarified

- Line 309: not clear from the text if reference (25) was irreproducible, or the other reference therein. The older reference should be cited for clarification.

Reference to previous paper added

Reviewer #2 (Remarks to the Author):

I am reviewer 2 from the previous round of reviews. The authors have made some useful improvements to the manuscript and fixed some important errors (the suggestion from reviewer 1 and fortunate discovery of the interfering SNP were the most important for this). I'm still unconvinced by the level of advance the paper provides (though see below for one possible exception), but I am willing to certify that it meets scientific soundness for the experiments and writing.

Previous review point 1: if the authors think that the knowledge provided in the paper opens up some avenue of approach or allows greater confidence for some sort of biosafety, I'd encourage them to describe a specific scenario (or scenarios) in the introduction and/or discussion in which more accurate knowledge of tract conversion length is critical to some sort of design or biosafety/regulatory aspect. Right now, I can only think of allelic drives designed to convert a tract with many mutations from a more limited number of gRNA target sites, and these seem to be more specific for pesticide resistance reduction in agricultural pests. In this case, the species-specific knowledge provided here

may still not be specific enough because I'm not sure if such drives are being considered for *Anopheles* mosquitoes.

We thank the reviewer for their approval of our article's rigour. This is the first real demonstration of the nature of resolution of homing-based drives and in our view – a view shared by the other reviewers – this is an important advance. The regulatory policy surrounding gene drives is still very much in development and running behind the scientific progress. We imagine policy makers may require empirical confirmation that *An. gambiae* gene drive releases would not induce runs of homozygosity, which this paper provides.

Perhaps the best way to frame this would be in the context of responding to a concern from a regulator or other key stakeholder about how much of the idiosyncratic release strain background would persist and spread during a gene drive release. Our data provides clearer answers to this concern; in addition, the relative rates of meiotic drive versus homing can impact this and are therefore an important advance in the literature.

While this is species-specific knowledge, there are multiple gene drives being developed in this species, which has been shown to be particularly amenable to this type of control. As we tested two gene drives with different promoters in different loci, these results will be applicable to multiple gene drives within *An. gambiae*.

The gesture of the reviewer to seek a second opinion from a member of their lab on the significance and soundness of the article was very magnanimous and fair; it is comforting to hear that this second opinion was positive.

Previous review point 2: I think I may not have been specific enough here. The introduction states that “A common concern is whether the lab-bred mosquito strains containing the gene drive will be capable of rapid transfer into wild populations due to lack of fitness after years of lab maintenance, slowing the initial spread of the gene drive element into the target population.” I was saying that the specific rates measured here aren't related to this question because even with the wider uncertainty range before this study narrow it down, it was still small enough for this to be only a tiny fraction of the genome and very unlikely to produce any fitness effects at all, much less significant effects.

This has been addressed in the previous comment.

Previous review point 3: Again, I'm having trouble seeing specific examples here, aside from the one specific case of allelic drive. As an aside, this specific case should probably be mentioned to try and increase the study's impact, though there may be some differences between conversion tracts when copying a large drive alleles versus just a small mutation.

An allelic drive application for these results, and applicability to toxin-antidote drive strategies, are discussed on Lines 343-351.

The response says that, “While it is true that a novel SNP introduced into a broken gene will be unlikely to have a phenotypic effect, it is still important to be able to predict repair outcomes on the large scale at which they would happen if gene drives were used in control strategies.”

I am having trouble seeing the importance of this. Maybe explain with an example?

We don't think an example is necessary in the main text as it would be unnecessarily speculative. In general, the results in this work are most targeted towards improving confidence in the predictable function of gene drives for stakeholders and policymakers. Worst-case scenarios are always discussed in this context, and therefore having empirical answers to worst-case scenario questions is important.

The response says that, “predictability of these repair outcomes is essential for the function of modification-based gene drives. This is particularly true for the conserved genes that all modern gene drives target, where single SNPs can have profound phenotypic effects.”

Maybe we are still discussing insecticide targeting drives, but the response here seems to indicate that it could be important for other drives. I don't think that all or even most rescue type modification drives that are essential for viability will have alleles that really need close tracking for this, even if they are the part of the gene or promoter region that's actually important. Again, the authors may be thinking of an example that I am not imagining that could strengthen their manuscript if it were included.

We think the relevance of this work to this question is that if it has been empirically proven that SNPs nearby to a gene drive site are not at risk of being driven to homozygosity, future applications of gene drive technology can be implemented with this in mind. For example, modification gene drives could be designed in loci with nearby SNPs that would be problematic at homozygosity, as it is known that these would not be transferred during homing.

For your response to my comment in line 126 and my new read of the paper, things seem to be getting more interesting. Apologies if I missed this before (though maybe it's just apparent in the new data for Figure 6), but this certainly seems to point to uneven inheritance of the two chromosomes. Meiotic drive is invoked as an explanation, which may be accurate. However, I think that a little bit more data is needed to support this, especially because the different batches seem to have different rates of donor

chromosome inheritance. It seems to me that in addition to attributing the difference to drive effects, there could be fitness differences between the chromosomes at sperm or offspring stages or even a natural meiotic drive. Unless I misunderstand the situation, it may be necessary to repeat the experiment (for Figure 6) with non-drive mosquitoes of the same hybrid strain and cross in order to be able to firmly attribute the difference to drive effects. If the drive does indeed turn out to be the cause, then it would be very interesting.

This is true, there may be fitness differences in sperm or offspring stages that can lead to meiotic drive, rather than it being solely a result of chromosome destruction – this has been added to lines 361-364. Future work would be necessary to confirm the mechanism of bias, but this is outside the scope of this paper.

One minor point I saw in the new version:

Line 172: “Error! Reference source not found.,”

This has been corrected

Because my opinion on this study’s impact seemed to be substantially different from the other reviewers, I found a junior colleague to provide a short review, which follows:

In this study, the authors investigated gene conversion tracts (GCTs) in *Anopheles* gene drive, demonstrating that around two-thirds of GCTs lengths were less than 50 base pairs while the rest was only a few hundred base pairs. They also observed the contribution of meiotic drive to biasing drive inheritance in certain tested pools. The revisions made in response to the previous round of review have improved the manuscript significantly. Overall, the manuscript is well-constructed, with reasonably designed experiments, solid data and clear writing. I only have a few concerns:

We would like to thank the fourth reviewer for agreeing to contribute to this process, and for their positive appraisal of our work.

To enhance the scientific impact of this study, it may be beneficial to further elucidate its broader application. In addition to the allelic drive mentioned by Reviewer 2, the findings of this study could potentially inform the design of toxin-antidote modification drives (<https://doi.org/10.1038/s41467-020-14960-3>). In such drives, modified SNPs in target sites are intended to be introduced into the genome via copying from homology arms and may be needed to prevent cleavage of the drive allele. However, if these modifications are situated more than a few hundred base pairs away, their transfer into the genome of *Anopheles* mosquitoes may prove challenging. More examples of potential application may also be discussed to strengthen the impact.

This is a good suggestion for applicability of the work and has been added (Lines 343-347).

Both short-read and long-read experiments supported the conclusion that meiotic drive existed in certain tested pools. While it is understood that multiple cuts in a chromosome (e.g., W-shredders) can disrupt the entire chromosome and induce meiotic drive, the mechanism by which a single cut achieves this is less clear. Providing additional explanations or references for this would enhance the comprehensibility of the manuscript.

Whether DSBs in one place on the chromosome would be enough to destroy the entire chromosome is still unproven; our work does appear to provide evidence to suggest this may be the case, perhaps through repeated cutting and microhomology-mediated repair. Alternatively, there may be a reduction in germ cell fitness caused by a similar mechanism, rather than outright chromosome destruction. We are not aware of any studies that have examined this for Cas9-mediated DSBs.

Typos:

Line 116: “ ...the length of donor sequenced copied into the recipient...” should be “...the length of donor sequence copied into the recipient...”.

Thank you, corrected

Line 172: “Error! Reference source not found”.

This has been corrected

Line 352: Imperial University?

This has been corrected to Imperial College London.

Reviewer #3 (Remarks to the Author):

In this manuscript, authors have undertaken to determine the extent to which gene conversion tracts (GCTs) are associated with CRISPR-induced cuts followed by homology-directed repair in *An. gambiae*. The authors show that most GCTs are short, thus facilitating rapid integration into the local genetic background.

The authors have now undertaken additional experiments to demonstrate more clearly that 1) long GCTs are not going undetected in their analyses and that 2) meiotic drive is the most likely explanation for disproportionate inheritance of the donor (drive) chromosome.

These additional studies have significantly improved the manuscript and hone in on greater accuracy in the results.

This work, to the best of my knowledge, presents the first systematic investigation of the mechanism of integration of gene drive elements from the donor chromosome into the recipient chromosome in *Anopheles*, and is therefore of significance to the field.

We would like to thank Reviewer 3 for their continuing support for the impact and validity of this work, and for their constructive comments to improve the manuscript.

Minor comments:

line 62: HEG gene

Fixed, thank you

Line 174: fix reference error

This has been fixed

Fig 4 legend: is there a better way to display the inaccurate proportion data? Is it possible to show a range or add a brief description of the degree of inaccuracy?

It's difficult to estimate the range or degree of inaccuracy in those samples, as none of the pools gave the expected ratio of donor to recipient chromosomes (due to the inheritance bias not being entirely due to homing of the gene drive). We think it's important to present the different haplotypes present in those samples, and only present the proportions for completeness. To make it clearer we have described the issue in lines 228-232 and reworked the description in Figure 3 and 4 legends.

REVIEWERS' COMMENTS

Reviewer #1 (Remarks to the Author):

I really like this third version, now with crystal clear explanations, figures nicely reworked (extra panel in figure 1, good rewording of figure legends and titles, frequencies based on more logical calculations), I'm happy to say that I now fully support publication.

Last polishing suggestions:

- There seems to be an error in Figure 3, pool A, left part of the graph, bottom bar Recipient + 86-268 bp : it looks more like Recipient + 0-10 bp .

- In the legend you could write “narrower bars” rather than “narrow rows”. Was it necessary to separate figures 3 and 4, which used to be single figure 3 in the previous version? Their legends are identical.

- Suppl information 1: in the legend of figs S2, S3 and S4 add (marked with an asterisk) after “can therefore be reasonably assumed to originate from that haplotype”. A similar addition is required in the legends of Fig S7-10. Delete “with” at the bottom of page 4. You may want to comment on the fact that in Fig S1, individual female samples, the two most common and real alleles are often found at strikingly unequal frequencies, and whether this could bias the allele frequency calculations in subsequent analyses (notably the quantification of meiotic drive).

Congratulations for all this work, which contributes invaluable fundamental knowledge on the intimate molecular processes of homing drives.